# Multi-head Temporal Latent Attention

**Keqi Deng, Philip C. Woodland**
Department of Engineering, University of Cambridge
Trumpington St., Cambridge, UK
kd502@cam.ac.uk, pw117@cam.ac.uk

## Abstract

While Transformer self-attention offers strong parallelism, the Key-Value (KV) cache grows linearly with sequence length and becomes a bottleneck for inference efficiency. Multi-head latent attention was recently developed to compress the KV cache into a low-rank latent space. This paper proposes Multi-head Temporal Latent Attention (MTLA), which further reduces the KV cache size along the temporal dimension, greatly lowering the memory footprint of self-attention inference. MTLA employs a hyper-network to dynamically merge temporally adjacent KV cache vectors. To address the mismatch between the compressed KV cache and processed sequence lengths, a stride-aware causal mask is proposed to ensure efficient parallel training and consistency with inference behaviour. Experiments across tasks, including speech translation, speech recognition, speech understanding and text summarisation, demonstrate that MTLA achieves competitive performance compared to standard Multi-Head Attention (MHA), while greatly improving inference speed and GPU memory usage. For example, on a English-German speech translation task, MTLA achieves a $5.3\times$ speedup and a reduction in GPU memory usage by a factor of 8.3 compared to MHA, while maintaining translation quality.

## 1 Introduction

The Transformer [44] decoder has become increasingly important, particularly with the success of large language models (LLMs) [7, 42]. As LLMs have been extended to other modalities such as speech [11, 40, 15], this decoder-only architecture is gradually becoming a unified framework for handling many tasks. For example, by placing an input speech sequence before the text and modelling causal dependencies auto-regressively via self-attention, decoder-only models can naturally handle speech tasks such as speech recognition and speech translation [47, 43]. However, during auto-regressive inference, each decoding step requires loading the cached attention keys and values to avoid re-encoding the history. This repeated memory access has emerged as a bottleneck, limiting inference speed and constraining both the decoding batch size and sequence length [38, 36, 22]. As model scales and application demands increase, reducing this memory bandwidth overhead is crucial for efficient deployment.

To alleviate the memory bottleneck associated with the Key-Value (KV) cache during incremental inference, several attention variants have been proposed. Multi-Query Attention (MQA) [38] reduces the number of KV heads by sharing a single head of keys and values across all query heads, greatly decreasing memory usage. Given that MQA can lead to quality degradation and training instability, Grouped-Query Attention (GQA) [1] was proposed, which partitions query heads into groups, each sharing a distinct head of keys and values. Despite these advancements, both MQA and GQA primarily focus on reducing the number of KV cache heads, which can lead to performance degradation due to limited representational capacity [30]. Recently, Multi-Head Latent Attention (MLA) [26] has emerged as a more advanced approach. MLA reduces the KV cache size by lowering the latent dimension of the saved KV vectors. [26, 30] show that MLA achieves higher model accuracy than

MQA and GQA, and can match or even surpass multi-head attention (MHA) [44]. However, existing methods, including MQA, GQA, and MLA, have not explored compression along the temporal dimension of the KV cache. Given that the KV cache size grows linearly with sequence length, there is great potential for further KV cache compression, especially in long-context scenarios.

This paper proposes Multi-Head Temporal Latent Attention (MTLA), which builds on MLA but further reduces the KV cache size along the temporal dimension. MTLA compresses the temporal dimension by dynamically merging temporally adjacent KV cache vectors in a learnable manner. Since the input sequence length varies across examples, this merging process cannot rely on static parameters: instead, MTLA employs a hyper-network to generate the merging weights for the KV cache. During inference the KV cache has fewer elements than the processed sequence and the most recent KV cache vectors can be updated as processing proceeds. However, the correct KV cache vectors must be used in training which is an issue for efficient parallel training. To address this issue, this paper designs a stride-aware causal mask to ensure consistency between the attention behaviour during parallel training and that during incremental inference. Following [26], decoupled rotary position embedding is adopted to encode positional information, together with MTLA temporal compression. Experiments on speech translation, speech recognition, speech understanding, and text summarisation show that MTLA achieves competitive model accuracy compared to standard MHA, while greatly improving inference speed and reducing GPU memory usage at inference.

The main contributions of this paper can be summarised in four main parts:

- MTLA is proposed, which is, to the best of our knowledge, the first self-attention mechanism capable of compressing the temporal dimension of the KV cache.

- A hyper-network is used to dynamically generate weights for merging adjacent KV caches along the temporal dimension.

- A stride-aware causal mask is designed for MTLA to achieve efficient parallel training, simulating the attention behaviour during incremental decoding.

- MTLA matches MHA and MLA in accuracy across tasks while greatly increasing processing speed and reducing GPU memory usage during inference. The code is fully open-sourced: `https://github.com/D-Keqi/mtla`

## 2 Related Work

Reducing the memory and computational overhead of the KV cache in Transformer decoders has been a focal point of recent research. MQA [38] reduces KV cache size by sharing a single key and value head across all query heads, while GQA [1] divides query heads into groups and each shares a single key and value head. MLA [26] compresses KV representations into a lower-dimensional latent space, offering better expressiveness than GQA and comparable or improved accuracy over MHA. Additionally, techniques like MiniCache [28] and MLKV [50] reduce memory by sharing KV caches across layers, though this may harm performance due to layer-specific attention patterns.

Another line of work explores linear attention models such as Linear Transformers [23, 45], RWKV [35], and Mamba [19], which reduce memory via linear time complexity. However, they often struggle with long-range dependencies, impacting tasks that rely on complex context. Recent theoretical analysis [2] also proves that truly subquadratic inference time can not solve challenging tasks such as document similarity. Despite the cost, quadratic attention remains crucial for fine-grained token interactions, motivating our focus on Transformer attention.

Beyond architectural modifications, various engineering techniques have been proposed to optimise Transformers. Dynamic token pruning methods, such as LazyLLM [17] and SnapKV [24], reduce memory usage by selectively removing less important tokens from the KV cache. [49] divides the context into chunks and inserts beacon tokens that store and accumulate information, effectively representing previous chunks to achieve context compression. Pruning can also be applied to attention heads or dimensions, though it may compromise contextual understanding and complicate the pipeline [30]. In addition, KV quantisation [27] can further reduce memory by lowering KV cache precision. Furthermore, FlashAttention [12, 13] restructures the attention computation to minimise memory access overhead, enhancing both speed and efficiency. While these tricks enhance Transformer efficiency, this paper focuses on directly compressing the KV cache along the temporal dimension, an under-explored direction that can greatly reduce memory and computation for long-sequence tasks.

[32] retrofits pre-trained LLMs by temporally compressing the KV cache, but cannot train from scratch and requires extra losses for each attention layer and head. In contrast, this work proposes a new attention mechanism requiring no changes beyond the attention module itself.

# 3    Preliminaries and Background

This section reviews some important background on the use of a KV-cache in auto-regressive inference and the operation of standard multi-head attention. The approaches taken by the MQA, GQA and MLA methods for reducing the size of the KV-cache are then outlined.

**Key-Value Cache in Auto-regressive Inference**    At inference, the model generates one next token $x_i$ at a time, using past tokens $x_1, \cdots, x_{i-1}$. To reduce computation, Transformers cache previously computed key and value vectors instead of re-computing the attention context for each step.

Given a query vector $\boldsymbol{q}_i \in \mathbb{R}^{1 \times d}$ at step $i$, where $d$ is the model dimension, and the cached key and value matrices $\mathbf{K}_{<i} \in \mathbb{R}^{(i-1) \times d}$ and $\mathbf{V}_{<i} \in \mathbb{R}^{(i-1) \times d}$, the attention output is computed as:

$$\text{Attention}(\boldsymbol{q}_i, \mathbf{K}_{<i}, \mathbf{V}_{<i}) = \text{softmax}\left(\frac{\boldsymbol{q}_i \mathbf{K}_{<i}^{\top}}{\sqrt{d}}\right) \mathbf{V}_{<i} \tag{1}$$

Here, $\boldsymbol{q}_i$ is computed from $x_i$, and $\mathbf{K}_{<i}, \mathbf{V}_{<i}$ are cached from previous steps. Without caching, $\mathbf{K}_{<i}$ and $\mathbf{V}_{<i}$ must be re-computed at every step, leading to redundant computation and quadratic time.

**Multi-Head Attention (MHA)**    Given an input sequence $\mathbf{X} \in \mathbb{R}^{T \times d}$, where $T$ denotes the sequence length, MHA [44] projects it into query $\mathbf{Q}$, key $\mathbf{K}$, and value tensors $\mathbf{V}$ using learned weight matrices:

$$\mathbf{Q} = \mathbf{X}\mathbf{W}_Q \in \mathbb{R}^{T \times (n_h \cdot d_h)}, \quad \mathbf{K} = \mathbf{X}\mathbf{W}_K \in \mathbb{R}^{T \times (n_h \cdot d_h)}, \quad \mathbf{V} = \mathbf{X}\mathbf{W}_V \in \mathbb{R}^{T \times (n_h \cdot d_h)} \tag{2}$$

where $\mathbf{W}_Q, \mathbf{W}_K, \mathbf{W}_V \in \mathbb{R}^{d \times (n_h \cdot d_h)}$ are learned matrices, and $n_h$ is the number of attention heads.

**Multi-Query Attention (MQA)**    MQA [38] shares key and value matrices across heads to reduce memory. Each head $h$ has its own query $\mathbf{Q}^{(h)} = \mathbf{X}\mathbf{W}_Q^{(h)} \in \mathbb{R}^{T \times d_h}$, but all heads share:

$$\mathbf{K} = \mathbf{X}\mathbf{W}_K \in \mathbb{R}^{T \times d_h}, \quad \mathbf{V} = \mathbf{X}\mathbf{W}_V \in \mathbb{R}^{T \times d_h} \tag{3}$$

**Group-Query Attention (GQA)**    GQA [1] groups heads into $g$ sets, each sharing a key and value.

$$\mathbf{K} = \mathbf{X}\mathbf{W}_K \in \mathbb{R}^{T \times (g \cdot d_h)}, \quad \mathbf{V} = \mathbf{X}\mathbf{W}_V \in \mathbb{R}^{T \times (g \cdot d_h)} \tag{4}$$

Heads in group $i$ share $\mathbf{K}^{(i)}, \mathbf{V}^{(i)} \in \mathbb{R}^{T \times d_h}$. Each head has independent queries as in MHA.

**Multi-Head Latent Attention (MLA)**    MLA [26] compresses the key-value memory into a latent sequence $\mathbf{C} \in \mathbb{R}^{T \times r}$ with a smaller hidden dimension $r < d$. The attention computation becomes:

$$\mathbf{C} = \mathbf{X}\mathbf{W}_r \in \mathbb{R}^{T \times r} \tag{5}$$

$$\mathbf{K} = \mathbf{C}\mathbf{W}_K \in \mathbb{R}^{T \times (n_h \cdot d_h)}, \quad \mathbf{V} = \mathbf{C}\mathbf{W}_V \in \mathbb{R}^{T \times (n_h \cdot d_h)} \tag{6}$$

where $\mathbf{C}$ is saved as KV cache and directly used for attention computation, avoiding explicit $\mathbf{K}$ and $\mathbf{V}$ computation by absorbing $\mathbf{W}_K$ into $\mathbf{W}_Q$ and $\mathbf{W}_V$ into the output projection.

# 4    Multi-head Temporal Latent Attention (MTLA)

This paper proposes MTLA, which, building upon compressing the Key-Value (KV) cache into a low-rank latent space as in MLA, further compresses the KV cache along the temporal dimension. Hence, MTLA can greatly reduce GPU memory usage and accelerate inference. Meanwhile, MTLA addresses the challenge of mismatched KV cache length and generated sequence length by introducing a stride-aware causal mask, enabling efficient parallel training.

As illustrated in Fig. 1, unlike conventional Multi-Head Attention (MHA) that maintains separate key and value cache vectors for each attention head, MTLA employs a shared low-rank latent vector

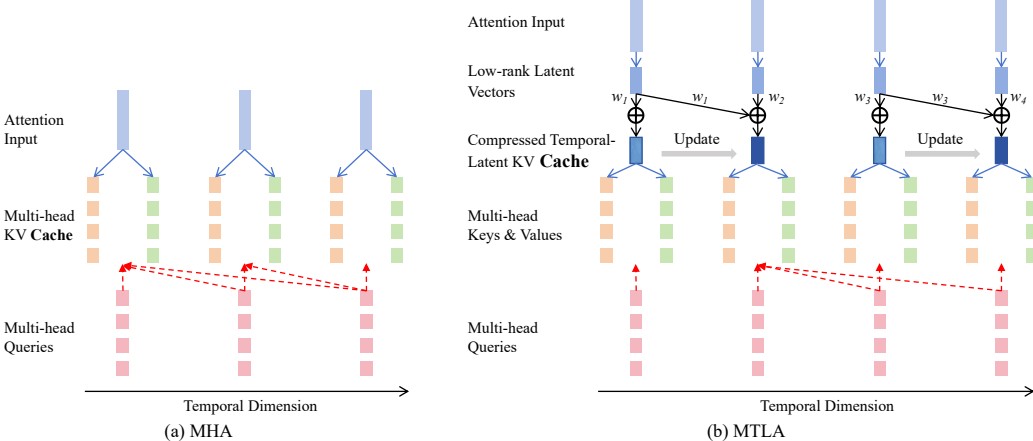

Figure 1: Illustration of MTLA. Blue arrows denote transformations by linear layers, and the red dashed lines indicate content attended to during attention. The example corresponds to 4 attention heads. (a) Standard MHA; (b) MTLA with a temporal compression ratio of 2. $\oplus$ denotes addition. The transformation from compressed temporal-latent KV cache to multi-head KVs can be absorbed into the query/output linear layers via matrix multiplication associativity, avoiding redundant computation.

to compress key and value information across heads, following [26]. Furthermore, MTLA merges adjacent latent vectors along the temporal dimension to store them as the KV cache.

Specifically, given an input sequence $\mathbf{X} \in \mathbb{R}^{T \times d}$, where $T$ is the sequence length and $d$ is the model dimension, the multi-head queries $\mathbf{Q} = (\boldsymbol{q}_1, \boldsymbol{q}_2, \cdots, \boldsymbol{q}_T)$ are computed following standard MHA:

$$\mathbf{Q} = \mathbf{X}\mathbf{W}_Q \in \mathbb{R}^{T \times (n_h \cdot d_h)} \tag{7}$$

where $\mathbf{W}_Q \in \mathbb{R}^{d \times (n_h \cdot d_h)}$ are learned linear weight matrices. Following [26], low-rank compression (dimension is $r$) is performed to obtain the low-rank latent vectors $\mathbf{C} = (\boldsymbol{c}_1, \boldsymbol{c}_2, \cdots, \boldsymbol{c}_T)$:

$$\mathbf{C} = \mathbf{X}\mathbf{W}_r \in \mathbb{R}^{T \times r} \tag{8}$$

where $\mathbf{W}_r \in \mathbb{R}^{d \times r}$ is a trainable weight matrix. Layer normalisation [4] is then applied to $\mathbf{C}$ to stabilise training, following the implementation in [26]. MTLA further applies learnable weights $(w_1, w_2, \ldots, w_T)$ to compress the latent sequence $\mathbf{C}$ along the temporal dimension, yielding a shorter compressed temporal-latent KV sequence $\hat{\mathbf{C}} = (\hat{\boldsymbol{c}}_1, \hat{\boldsymbol{c}}_2, \cdots, \hat{\boldsymbol{c}}_t) \in \mathbb{R}^{t \times r}$, where $t = \lceil T/s \rceil$ and $s$ denotes the temporal compression ratio.

As illustrated in Fig. 1, assuming $s = 2$, every 2 temporally adjacent latent vectors in $\mathbf{C}$ are merged using the corresponding weights $(w_1, w_2, \ldots, w_T)$; for example, $\hat{\boldsymbol{c}}_1 = w_1 \cdot \boldsymbol{c}_1 + w_2 \cdot \boldsymbol{c}_2$, and $\hat{\boldsymbol{c}}_2 = w_3 \cdot \boldsymbol{c}_3 + w_4 \cdot \boldsymbol{c}_4$. Since the length of $(w_1, w_2, \ldots, w_T)$ varies dynamically with the input and cannot be handled using static parameters, MTLA utilises a hyper-network that takes $\mathbf{C}$ as input to generate $(w_1, w_2, \ldots, w_T)$. Further details of this hyper-network are given in refer to Sections 4.1 and 4.2. The choice of $s$ effectively controls the extent of KV cache compression in MTLA. However, choosing too large a value can caused marked performance degradation.

With the cached $\hat{\mathbf{C}} \in \mathbb{R}^{t \times r}$, the keys $\mathbf{K}$ and values $\mathbf{V}$ can be obtained through up-projection matrices and used for attention computation:

$$\mathbf{K} = \hat{\mathbf{C}}\mathbf{W}_K \in \mathbb{R}^{t \times (n_h \cdot d_h)}, \tag{9}$$

$$\mathbf{V} = \hat{\mathbf{C}}\mathbf{W}_V \in \mathbb{R}^{t \times (n_h \cdot d_h)}, \tag{10}$$

$$\mathbf{Y} = \text{softmax}\left(\frac{\mathbf{Q}\mathbf{K}^\top}{\sqrt{d_h}}\right)\mathbf{V}\mathbf{W}_O \in \mathbb{R}^{T \times d} \tag{11}$$

where $\mathbf{W}_K, \mathbf{W}_V \in \mathbb{R}^{r \times (n_h \cdot d_h)}$, and $\mathbf{W}_O \in \mathbb{R}^{(n_h \cdot d_h) \times d}$ are are learned linear weight matrices. Note that due to the associative property of matrix multiplication, Eq. 11 can be rewritten as:

$$\text{softmax}\left(\frac{\mathbf{Q}\mathbf{K}^\top}{\sqrt{d_h}}\right)\mathbf{V}\mathbf{W}_O = \text{softmax}\left(\frac{\mathbf{X}(\mathbf{W}_Q\mathbf{W}_K{}^\top)\hat{\mathbf{C}}^\top}{\sqrt{d_h}}\right)\hat{\mathbf{C}}(\mathbf{W}_V\mathbf{W}_O) \tag{12}$$

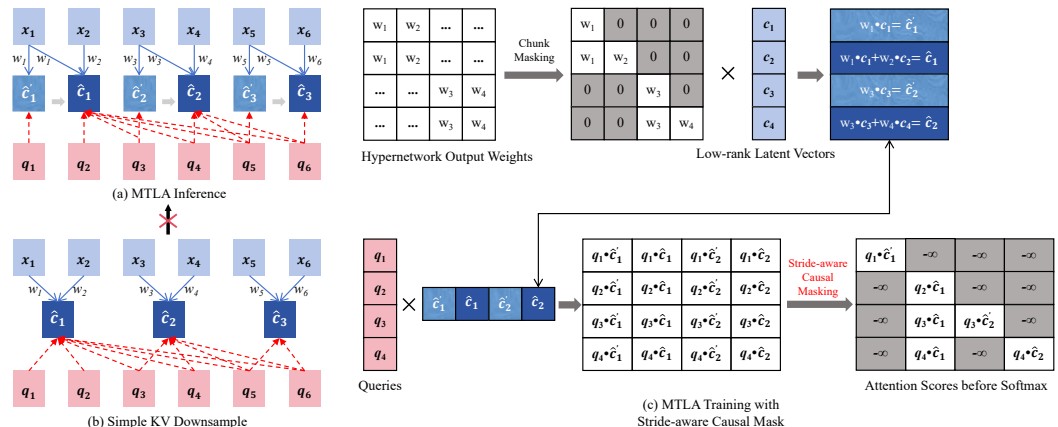

Figure 2: Illustration of MTLA inference and training with temporal compression ratio 2. $\boldsymbol{q}_i$: query, $\boldsymbol{x}_i$: attention input, $\hat{\boldsymbol{c}}_j$: compressed KV cache, $\hat{\boldsymbol{c}}_j'$: temporary version updated later. (a) Incremental inference in MTLA, where at certain steps (e.g., 1, 3, 5), the model attends to the temporary $\hat{\boldsymbol{c}}_j'$. (b) KV cache generated by simple pre-downsampling, which mismatches MTLA inference. (c) MTLA training, where a stride-aware causal mask is used to match the inference condition.

Therefore, the cached $\hat{\mathbf{C}}$ can be directly used for attention computation without explicitly computing the keys and values, as $\mathbf{W}_K$ and $\mathbf{W}_V$ can be absorbed into $\mathbf{W}_Q$ and $\mathbf{W}_O$, respectively.

## 4.1 Inference using MTLA

Fig. 2(a) illustrates inference using MTLA. Specifically, given a new input vector $\boldsymbol{x}_i$, the corresponding low-rank latent vector $\boldsymbol{c}_i$ is first computed following Eq. 8. Then, $\boldsymbol{c}_i$ is fed into the hyper-network to generate the corresponding weight $w_i$. Specifically, the weight is computed as follows:

$$w_i = \text{Sigmoid}\left(\text{Linear}(\boldsymbol{c}_i) \cdot \text{Linear}(\boldsymbol{pe}_j)\right) \tag{13}$$

where $j = \lceil i/s \rceil$, Linear denotes a linear layer transformation, $\boldsymbol{pe}_j$ is the positional embedding at step $j$ [44], and $\cdot$ denotes element-wise multiplication.

Once $w_i$ is obtained, the compressed temporal-latent KV cache can be updated. If the remainder of $i/s$ equals 1 (assuming $i$ starts from 1), the cache is updated as $\hat{\mathbf{C}} = \text{Concat}(\hat{\mathbf{C}}, w_i \boldsymbol{c}_i)$ where Concat denote concatenation; otherwise, the $j$-th cache vector is updated as $\hat{\boldsymbol{c}}_j = \hat{\boldsymbol{c}}_j + w_i \boldsymbol{c}_i$. Note that until the remainder of $i/s$ equals 0, each $\hat{\boldsymbol{c}}_j$ here actually corresponds to $\hat{\boldsymbol{c}}_j'$ in Fig. 2, which will be updated in later steps. Then, the attention output is computed following Eq. 12.

## 4.2 MTLA Training with Stride-aware Causal Mask

As shown in Fig. 2(a), during inference, queries at certain steps attend to the temporary $\hat{\boldsymbol{c}}_j'$. As shown in Fig. 2(b), simply using pre-downsampling to obtain compressed KV vectors for attention computation during training fails to match inference behaviour. Therefore, enabling efficient parallel training poses a challenge. This paper proposes a stride-aware causal mask to address this issue.

During training, as shown in Fig. 2(c), MTLA computes the compressed temporal-latent KV sequence as:

$$\hat{\mathbf{C}}' = (\underbrace{\hat{\boldsymbol{c}}_1', \ldots, \hat{\boldsymbol{c}}_1}_{s}, \cdots, \underbrace{\hat{\boldsymbol{c}}_t', \ldots, \hat{\boldsymbol{c}}_t}_{s}) \tag{14}$$

where $s$ is the temporal compression ratio and $t = \lceil T/s \rceil$. Therefore, this sequence length remains $T$ (only in training). To compute the sequence $\hat{\mathbf{C}}'$, the compressed low-rank latent vectors $\mathbf{C}$ are first passed through a hyper-network. To ensure parallel training efficiency, MTLA computes $\hat{\mathbf{C}}'$ using matrix multiplication. Specifically, the hyper-network generates a weight matrix based on the input

C:

$$\mathbf{PE} = (\underbrace{\boldsymbol{pe}_1, \ldots, \boldsymbol{pe}_1}_{s}, \cdots, \underbrace{\boldsymbol{pe}_t, \ldots, \boldsymbol{pe}_t}_{s}) \tag{15}$$

$$\mathbf{W} = \mathrm{Sigmoid}(\mathrm{Linear}(\mathbf{PE}) \times \mathrm{Linear}(\mathbf{C})) \in \mathbb{R}^{T \times T} \tag{16}$$

where $\mathbf{PE}$ consists of the replicated positional embedding vectors $\boldsymbol{pe}_j$ and $\times$ denotes matrix multiplication. As shown in the upper part of Fig. 2(c), after applying chunk masking (commonly used in streaming Transformer encoders [10]) to the resulting $\mathbf{W}$, it is multiplied with $\mathbf{C}$ to obtain $\hat{\mathbf{C}}'$.

The resulting $\hat{\mathbf{C}}'$ is then used for attention computation as in Eq. 12 (serving as $\hat{\mathbf{C}}$ in Eq. 12). However, instead of using a standard causal mask to prevent access to future information before the softmax, a stride-aware causal mask is proposed, as shown in the lower part of Fig. 2(c), to match the attention pattern of MTLA during incremental inference. Specifically, let $m$ denote the row index and $n$ the column index; the stride-aware causal mask is zero only when $n = m$ or $n < m$ and $n \bmod s = 0$, and $-\infty$ elsewhere. With this stride-aware causal mask, MTLA training retains the parallel efficiency of standard attention.

### 4.3 Decoupled Rotary Position Embedding in MTLA

If Rotary Position Embedding (RoPE) [39] is to be used, similar to MLA [26], MTLA also requires the use of decoupled RoPE [26]. A simple method is proposed in this paper to compress the cached keys of decoupled RoPE along the temporal dimension. Specifically, the queries obtained from Eq. 7 are rotated with a position-dependent matrix to produce RoPE queries $\mathbf{Q}^R = (\boldsymbol{q}_1^R, \boldsymbol{q}_2^R, \cdots, \boldsymbol{q}_T^R) \in \mathbb{R}^{T \times (n_h \cdot d_h^R)}$, where $d_h^R$ denotes per-head dimension for the decoupled RoPE. Similarly, the keys can also be obtained as in Eq. 9 and rotated with a position-dependent matrix to obtain RoPE keys $\mathbf{K}^R = (\boldsymbol{k}_1^R, \boldsymbol{k}_2^R, \cdots, \boldsymbol{k}_T^R) \in \mathbb{R}^{T \times d_h^R}$.

Next, $\mathbf{K}^R$ is compressed along the temporal dimension to obtain $\hat{\mathbf{K}}^R = (\hat{\boldsymbol{k}}_1^R, \hat{\boldsymbol{k}}_2^R, \cdots, \hat{\boldsymbol{k}}_t^R) \in \mathbb{R}^{t \times d_h^R}$. At inference, the most recent element in the RoPE key cache $\hat{\mathbf{K}}^R$ can also be updated. If the remainder of $i/s$ equals 1, this cache is updated as $\hat{\mathbf{K}}^R = \mathrm{Concat}(\hat{\mathbf{K}}^R, \boldsymbol{k}_i^R)$; otherwise, the $j$-th cache vector is updated as $\hat{\boldsymbol{k}}_j^R = \boldsymbol{k}_i^R$. Then, the RoPE query-key pairs are used to augment the attention computation and Eq. 11 and Eq. 12 can be rewritten as:

$$\mathbf{Y} = \mathrm{softmax}\left( \frac{\mathbf{X}(\mathbf{W}_Q \mathbf{W}_K^\top)\hat{\mathbf{C}}^\top + \mathbf{Q}^R(\hat{\mathbf{K}}^R)^\top}{\sqrt{d_h}} \right) \hat{\mathbf{C}}(\mathbf{W}_V \mathbf{W}_O) \tag{17}$$

where $\mathbf{X} \in \mathbb{R}^{1 \times d}$ in incremental inference, and when multiplying $\mathbf{Q}^R \in \mathbb{R}^{T \times (n_h \cdot d_h^R)}$ with $(\hat{\mathbf{K}}^R)^\top \in \mathbb{R}^{d_h^R \times T}$, the head number of keys must first be repeated, following MQA [38].

This design of compressing decoupled RoPE keys along the temporal dimension simplifies the training process: based on Eq. 17, the original $\mathbf{K}^R \in \mathbb{R}^{T \times d_h^R}$ can be directly used in place of $\hat{\mathbf{K}}^R$ (also using $\hat{\mathbf{C}}'$ instead of $\hat{\mathbf{C}}$ as mentioned in Section 4.2), and the attention output can be computed with the proposed stride-aware causal mask.

Assuming the number of self-attention layers is $l$, then for standard MHA, each token corresponds to $2d_h n_h l$ elements in the KV cache. For MTLA, for simplicity, this paper follows the hyper-parameter settings of [26], setting $r = 4d_h$ and $d_h^R = d_h/2$. Therefore, the average number of KV cache elements per token in MTLA is $9d_h l/(2s)$. The default value of $s$ is set to 2, making $9d_h l/(2s) = 2.25d_h l$ close to the KV cache elements per token in MQA (i.e. $2d_h l$).

## 5 Experimental Setup

In this section, the proposed MTLA approach is evaluated on a range of tasks, including speech translation (ST), text summarisation, automatic speech recognition (ASR), and spoken language understanding (SLU), and is compared with standard MHA and advanced MLA. Since this work focuses on self-attention, the experiments are conducted using a Transformer-based decoder-only architecture, implemented within the Fairseq [33] toolkit.

### 5.1 Datasets

The ST task uses the MuST-C [16] v1.0 English-German (En-De) dataset, with data preprocessing following the Fairseq example. The text summarisation task is conducted on the XSum [31] dataset. For the ASR task, the AMI [8] dataset is employed. For the SLU task, the SLURP [5] dataset is used to evaluate intent classification. More details of the datasets used are given in Appendix C.

### 5.2 Model Specifications

Since this paper focuses on self-attention, the model is built based on a Transformer decoder, where the encoder output is prepended to the input of the self-attention module as a prompt, and the cross-attention module is removed. This is sometimes referred to as a decoder-only structure. As a result, the cached keys and values will contain information from the encoder output. The proposed MTLA, along with the standard MHA and the MLA technique, are each used as the self-attention module to build the model, while all other components are kept strictly identical. In the following sections, the overall models built with MTLA, MHA, and MLA self-attention modules are referred to as MTLA, MHA, and MLA for simplicity.

The decoder used for all tasks shares the same configuration with 512 attention dimensions and 8 heads. For MTLA and MLA, $r$ in Eq. 8 is set to 256 and $d_h^R$ is set to 32. In MTLA, the temporal compression rate $s$ is set to 2 by default unless otherwise specified. For the ST task, following the Fairseq example, a Transformer encoder is used and initialised with ASR task weights. For the text summarisation task, a standard Transformer encoder is used. For the ASR task, a Transformer encoder is employed. For the SLU task, a Conformer [20] encoder is used. More details can be found in Appendix D.

### 5.3 Metrics

All inference speed tests are conducted on the same NVidia RTX 6000 Ada GPU. To ensure a fair comparison, all models used the same batch size and beam size during inference. Inference time and the average GPU memory usage during inference are reported to evaluate efficiency. For the ST task, case-sensitive detokenized BLEU [34] is reported. For the text summarisation task, ROUGE [25] is used to evaluate summarisation quality, and ROUGE-1, ROUGE-2 (unigram and bigram overlap), and ROUGE-L (longest common subsequence) scores are reported. For speech recognition, word error rate (WER) results are reported. For the SLU task, accuracy is used to measure intent classification (IC).

## 6 Experimental results

This paper evaluates the proposed MLTA across tasks, including ST, text summarisation, ASR, and SLU, as both speech sequences and document texts are long sequences. Due to our computational resource constraints that make large-scale pre-training infeasible, all experiments are conducted using decoder-only architectures trained from scratch, allowing the effectiveness of MTLA to be assessed. To ensure reproducibility, this paper builds upon standard open-source implementations, such as the Transformer-based ST example in Fairseq. The goal is not to pursue task-specific state-of-the-art results, but to systematically compare MTLA with MHA and MLA under consistent and general model configurations. For each task, representative published results are reported to provide context. Appendix E presents MTLA's performance on the LRA benchmark [41], while Appendix F provides machine translation results to further evaluate MTLA on tasks involving relatively shorter sequences.

### 6.1 ST Task Results

The ST results are shown in Table 1. Overall, the models built in this paper achieve competitive performance on the MuST-C En-De benchmark dataset. The published results listed in Table 1 also use Transformer models, but based on an encoder-decoder architecture with cross-attention. Table 1 results show that our built decoder-only architecture can achieve similar performance with the same data and model scale. Comparing MHA and MLA, it is clear that MLA performs well: MLA results in only a limited reduction in translation quality drop (by 0.19 BLEU points) and offers improved inference speed and memory efficiency compared to MHA. Building upon MLA,

Table 1: BLEU (↑) results on the MuST-C En-De tst-COMMON set for multi-head attention (MHA), multi-head latent attention (MLA), and multi-head temporal latent attention (MTLA). ESPnet-ST [21] published results are broadly comparable (same data/scale; minor implementation differences).

| ST Model | Quality (BLEU) | Inference Time (s) | Speedup | Inference GPU Memory (MiB) | |
|---|---|---|---|---|---|
| | | | | Avg. Usage | Reduction Factor |
| ESPnet-ST [21] | 22.9 | — | — | — | — |
| MHA | 23.18 | 281.3 | 1.00× | 18646 | 1.00 |
| MLA | 22.97 | 97.0 | 2.90× | 5065 | 3.68 |
| Proposed MTLA | 23.28 | 65.6 | 4.29× | 2835 | 6.58 |
| Proposed MTLA w/ $s = 3$ | 23.25 | 52.7 | 5.34× | 2251 | 8.28 |
| Proposed MTLA w/ $s = 4$ | 23.05 | 48.7 | 5.78× | 1921 | 9.71 |

our proposed MTLA further improves the efficiency of the attention mechanism. With the default temporal compression ratio (i.e., 2), MTLA even slightly outperforms MHA in translation quality, suggesting that compressing redundant historical KV information may sometimes benefit model performance. Compared to MHA, MTLA achieves 4.29× speedup in inference and reduces average GPU memory consumption by a factor of 6.58.

Assuming the sequence length is $T$, MTLA reduces the per-token computational complexity during decoding from $\mathcal{O}(T)$ to $\mathcal{O}(T/s)$. Since self-attention is not the only component in the model (e.g., feed-forward networks also contribute), setting $s = 2$ does not directly halve the inference time. Moreover, the reported GPU memory usage includes both activation memory and the storage of KV Cache, so memory consumption is not halved either. Nevertheless, setting $s = 2$ already yields substantial efficiency gains: MTLA achieves a 1.48× speedup in overall inference and reduces overall GPU memory consumption by 1.79× compared to MLA. These gains become even more substantial with larger $s$. For instance, with $s = 4$, GPU memory usage is reduced by 2.64×.

## 6.2 Results on Other Tasks

Table 2: ROUGE (↑) results on the XSum test set. ROUGE-1 (R1) (↑), ROUGE-2 (R2) (↑), and ROUGE-L (RL) F1 (↑) scores are reported. The published result of TransformerABS [29] is broadly comparable to our results.

| Model | R1 | R2 | RL | Inference Time (s) | Speedup | Inference GPU Memory (MiB) | |
|---|---|---|---|---|---|---|---|
| | | | | | | Avg. Usage | Reduction Factor |
| TransformerABS [29] | 29.41 | 9.77 | 23.01 | — | — | — | — |
| MHA | 28.83 | 9.67 | 23.33 | 352.3 | 1.00× | 16141 | 1.00 |
| MLA | 29.39 | 9.87 | 23.78 | 141.1 | 2.50× | 3746 | 4.30 |
| Proposed MTLA | 29.14 | 9.79 | 23.60 | 105.2 | 3.35× | 2198 | 7.34 |

Table 3: WER (↓) results on the AMI IHM test set for MHA, MLA, and the proposed MTLA. ESPnet published [46] results are listed but not directly comparable to our built models.

| ASR Model | WER | Inference Time (s) | Speedup | Inference GPU Memory (MiB) | |
|---|---|---|---|---|---|
| | | | | Avg. Usage | Reduction Factor |
| ESPnet [46] | 16.49 | — | — | — | — |
| MHA | 12.98 | 269.4 | 1.00× | 17509 | 1.00 |
| MLA | 12.67 | 105.3 | 2.56× | 4415 | 3.97 |
| Proposed MTLA | 12.66 | 71.8 | 3.75× | 2364 | 7.41 |

Experiment conclusions across text summarisation, ASR, and SLU tasks (Tables 2, 3, and 4) are generally consistent with those from the ST experiments. First, our built models achieve competitive performance across different tasks. Second, compared to MHA, MLA achieves competitive accuracy (ROUGE scores, WER, and IC accuracy) and better inference efficiency. Our proposed MTLA further improves inference efficiency. Compared to MHA, MTLA achieves up to 3.75× speedup and

Table 4: Accuracy (↑) results of intent classification (IC) on the SLURP test set for MHA, MLA, and the proposed MTLA. ESPnet-SLU [3] published result is generally comparable to our built models.

| SLU Model | Accuracy | Inference Time (s) | Speedup | Inference GPU Memory (MiB) | |
|---|---|---|---|---|---|
| | | | | Avg. Usage | Reduction Factor |
| ESPnet-SLU [3] | 86.3 | — | — | — | — |
| MHA | 86.83 | 133.1 | 1.00× | 14370 | 1.00 |
| MLA | 86.93 | 61.2 | 2.17× | 3343 | 4.30 |
| Proposed MTLA | 86.80 | 52.7 | 2.53× | 2051 | 7.01 |

reductions in GPU memory use by more than a factor of 7, while maintaining or even improving task performance. These results highlight the broad applicability and practical benefits of our decoder-only architecture and MTLA KV cache compression method across various sequence tasks.

## 6.3 Comparisons with Related Work

Table 5: BLEU (↑) results on the MuST-C En-De tst-COMMON set for related methods, including Multi-Query Attention (MQA) and Group-Query Attention (GQA) with a group size of 2.

| ST Model | Quality (BLEU) | Inference Time (s) | Speedup | Inference GPU Memory (MiB) | |
|---|---|---|---|---|---|
| | | | | Avg. Usage | Reduction Factor |
| MHA | 23.18 | 281.3 | 1.00× | 18646 | 1.00 |
| MQA | 22.70 | 168.1 | 1.67× | 3074 | 6.07 |
| GQA | 22.75 | 190.6 | 1.48× | 5313 | 3.51 |
| MLA | 22.97 | 97.0 | 2.90× | 5065 | 3.68 |
| MLA w/ SnapKV [24] | 21.76 | 80.8 | 3.48× | 4222 | 4.42 |
| Mamba-2 [14] | 18.62 | 157.5 | 1.78× | 5676 | 3.29 |
| Proposed MTLA | 23.28 | 65.6 | 4.29× | 2835 | 6.58 |
| Proposed MTLA w/ $s = 3$ | 23.25 | 52.7 | 5.34× | 2251 | 8.28 |
| Proposed MTLA w/ $s = 4$ | 23.05 | 48.7 | 5.78× | 1921 | 9.71 |

This subsection further compares our work with other approaches, including MQA and GQA. First, MLA and our MTLA follow the hyper-parameter settings of [26], as discussed in Section 4.3. Under this configuration, each token in MLA results in a KV cache size equivalent to that of GQA with 2.25 groups. Therefore, the GPU memory usage for inference is similar between MLA and GQA. Note that the GPU memory usage reported here includes both intermediate activations and the KV cache.

Importantly, MLA achieves faster inference than GQA and also outperforms MQA in speed, demonstrating that storing KV information in low-rank latent vectors and directly using them in attention reduces computation accelerates inference. Moreover, MLA also outperforms GQA in translation quality, which is why this paper focuses comparisons to it.

For our proposed MTLA, with the default temporal compression rate $s = 2$, its pre-token KV cache elements are equivalent to GQA with $2.25/2 = 1.125$ groups. Since MQA corresponds to GQA with 1 group, the KV cache size of MTLA becomes roughly equivalent to that of MQA. This motivates our choice of $s = 2$ as the default setting. As shown in Table. 5, MTLA yields similar memory usage as MQA while delivering 2.56× inference speedup. This is because MTLA inherits the low-rank compression benefits of MLA and further reduces per-token complexity from $\mathcal{O}(T)$ to $\mathcal{O}(T/s)$, with $T$ as the sequence length. In contrast, MQA and GQA offer limited speedups over MHA and mainly reduce GPU memory usage.

As noted in Sec. 5.3, all inference speed tests use the same batch and beam size across models. MTLA is a more advanced KV compression method than MQA (i.e., GQA with 1 group), which cannot reduce group count further, while MTLA allows further compression by increasing $s$. For example, with $s = 4$, MTLA significantly outperforms MQA in translation quality ($p < 0.05$, statistically tested via SacreBLEU [37]), while also yielding greater inference speed and GPU memory reduction.

This section further applies SnapKV [24], a representative token compression method, to MLA for comparison with MTLA. The results in Table. 5 show that while MLA with SnapKV improves

inference efficiency compared to MLA, it also leads to some reduction in translation quality. MTLA outperforms MLA with SnapKV in terms of quality, inference time, and GPU memory usage. The speech translation task is a strong test of whether sufficient information can be preserved when compressing tokens or context, and the results demonstrate that MTLA excels in this regard.

As a further point of comparison, this section implements Mamba-2 [14] and compares it to MTLA. The results in Table. 5 show that MTLA outperforms Mamba-2 in inference efficiency on this task and also on translation quality. While linear-complexity models like Mamba-2 will certainly yield more efficient inference than quadratic attention mechanisms when dealing with extremely long sequences, the model performance can also suffer. In summary, MTLA follows the mainstream approach of using quadratic attention mechanisms and therefore benefits from stronger model performance while greatly improving inference time and GPU memory usage.

### 6.4 Extended Results with FlashAttention-2

This section further employs FlashAttention-2 [12] to evaluate MTLA under a stronger inference implementation. Since the official FlashAttention-2 does not directly support MTLA, this paper extends it by implementing custom CUDA kernels for MTLA inference[1]. As shown in Table 6, while using FlashAttention-2 certainly accelerates inference, it does not change the conclusion, as MTLA still achieves a 3.99× speedup in inference and reduces average GPU memory consumption by a factor of 7.34 compared to MHA.

Table 6: BLEU (↑) results on the MuST-C En-De tst-COMMON set, with or without FlashAttention-2.

| ST Model | Quality (BLEU) | Inference Time (s) | Speedup | Inference GPU Memory (MiB) | |
|---|---|---|---|---|---|
| | | | | Avg. Usage | Reduction Factor |
| MHA | 23.18 | 281.3 | 1.00× | 18646 | 1.00 |
| w/ FlashAttention-2 | 23.16 | 145.7 | 1.93× | 9244 | 2.02 |
| Proposed MTLA | 23.28 | 65.6 | 4.29× | 2835 | 6.58 |
| w/ extended FlashAttention-2 | 23.29 | 36.5 | 7.71× | 1259 | 14.81 |

## 7  Conclusions

This paper proposes MTLA, the first self-attention mechanism capable of compressing the temporal dimension of the KV cache. Building upon the low-rank KV compression of MLA, MTLA employs a hyper-network to dynamically merge adjacent KV caches, enabling effective temporal compression. A stride-aware causal mask is proposed to ensure that MTLA maintains efficient parallel training while matching the attention behaviour during incremental inference, addressing the mismatch between the compressed KV cache length and the processed sequence length. Experiments across ST, text summarisation, ASR, and SLU show that MTLA greatly accelerates inference and reduces GPU memory usage at inference without sacrificing accuracy. With a temporal compression rate of 2, MTLA already matches the KV cache compression level of MQA while delivering better accuracy and speed, and it supports further compression, establishing itself as a more advanced KV cache compression method. Further comparisons show that MTLA consistently outperforms MLA with SnapKV and the linear Mamba-2 model in both quality and efficiency. Even with FlashAttention-2 acceleration, MTLA maintains up to 3.99× faster inference and 7.34× lower memory usage than MHA, confirming its effectiveness as a general and advanced KV cache compression approach. Future work will explore applying MTLA to LLMs, where the KV cache size is a key bottleneck during inference. MTLA temporal compression of the KV cache offers a promising way to scale LLMs to longer contexts while balancing memory use, latency, and accuracy.

## Acknowledgments

Keqi Deng is funded by the Cambridge Trust.

---

[1]The specific implementation refers to `https://github.com/D-Keqi/mtla`

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

## A  Limitations

Due to limited computational resources, this work does not investigate large language model (LLM) pre-training. The proposed MTLA is designed specifically for decoder-only architectures and can efficiently compress the KV cache. Standard text-based LLMs are successful examples of decoder-only models. Recent studies have shown that pre-pending speech representations as prompts before the self-attention input can extend text-based LLMs to speech tasks. However, building an LLM based on MTLA or replacing self-attention in a pre-trained LLM with MTLA and re-training it requires very substantial computational resources, which we do not possess. As a result, we are unable to construct an LLM based on MTLA to verify its extension to other tasks, such as speech. Instead, we construct decoder-only models and train them from scratch to evaluate MTLA across a range of tasks.

Second, as Transformer-based models have been extensively developed by the community in recent years, there is a large amount of related work. It is not feasible for us to implement and compare all such approaches. In this work, we compare MTLA with the most relevant and representative KV-cache compression methods, including MQA, GQA, and MLA. In addition, we also include comparisons with the typical token compression method SnapKV and the state space model Mamba-2. Further comparisons are only feasible through theoretical discussion, as presented in Section 2.

This work focuses on long-sequence tasks, particularly speech, due to the naturally long sequence length of speech inputs. We also conduct evaluations on a text summarisation task. While many additional tasks could be used to further evaluate its effectiveness of MTLA, we leave such investigations for future work. Given the growing dimensionality of modern LLMs and the increasing use of long reasoning chains to improve output quality, the MTLA, which compresses the KV cache along both the latent and temporal dimensions, can be particularly valuable.

## B  Broader impact

Decoder-only architectures based on self-attention have become increasingly popular in recent years, especially in the context of large language models (LLMs). However, due to their high dimensionality and massive number of parameters, LLMs incur expensive inference costs and are heavily dependent on GPUs. This problem is further exacerbated by the use of chain-of-thought, which enhances reasoning ability but results in significantly longer output sequences, making inference even more costly. Such inference consumes substantial energy from GPUs. By contrast, our proposed MTLA compresses the Key-Value Cache in both latent and temporal dimensions, greatly improving inference efficiency, which can be of great value to make LLMs more energy-efficient and environmentally sustainable. Therefore, our work has the potential to generate a positive societal impact. We do not know of any negative societal impact.

## C  Data Set Statistics

The ST task uses the MuST-C [16] v1.0 English-German (En-De) dataset, with data preprocessing following the Fairseq example, using 8,000 unigrams as the target language modelling units and fbank features as input. The text summarisation task is conducted on the XSum [31] dataset, where 30,000 BPE units are used. For the ASR task, the AMI [8] dataset is employed. Due to the challenging nature of the data, fixed WavLM [9] Large features are extracted using the S3PRL [48] toolkit as input. When measuring inference speed, this feature is pre-stored and 100 BPE units are used. For the SLU task, the SLURP [5] dataset is used to evaluate intent classification, with fbank features as input. Following [3], intent classification is performed by jointly predicting the transcription and the intent to achieve better performance. A total of 500 BPE units are used for transcription modelling.

The data set statistics for the datasets used in the experiments are shown in Table 7. The MuST-C [16] v1.0 En-De dataset comprises English-German speech translation data collected from TED Talks. The Augmented Multi-Party Interaction (AMI) Meeting Corpus [8] offers 100 hours of English meeting recordings captured in instrumented rooms, featuring multimodal data such as audio, video, and whiteboard content, with annotations including speech transcriptions and dialogue acts. The Spoken Language Understanding Resource Package (SLURP) [5] dataset is a comprehensive English spoken language understanding resource encompassing 18 domains, designed to facilitate tasks like intent

classification and slot filling, with a diverse set of utterances. The XSum [31] dataset consists of BBC news articles from 2010 to 2017, each paired with a single-sentence abstractive summary, totalling over 226K document-summary pairs, and is widely used for evaluating summarisation models.

Table 7: Statistics of datasets used in this paper

| | MuST-C v1.0 En-De | |
|---|---|---|
| Domain | TED Talk | |
| Train set | train | |
| -Duration | 400.0 hours | |
| -German words | 3880K | |
| Test sets | dev | tst-COMMON |
| -Duration | 2.3 hours | 4.1 hours |
| -German words | 26K | 44K |
| | XSum Dataset | |
| Domain | BBC News Articles | |
| Train set | train | |
| -Documents | 204K | |
| -Avg. article length | 431 words | |
| -Avg. summary length | 23 words | |
| Test sets | dev | test |
| -Documents | 11K | 11K |
| | AMI Meeting Corpus | |
| Domain | Meetings | |
| Train set | train | |
| -Duration | 100.0 hours | |
| -Utterances | 108K | |
| Test sets | dev | test |
| -Utterances | 13K | 12K |
| | SLURP Dataset | |
| Domain | Human-Computer Interaction (HCI) commands | |
| Train set | train | |
| -Duration | 83.7 hours | |
| -Utterances | 120K | |
| Test sets | dev | test |
| -Duration | 6.9 hours | 10.3 hours |
| -Utterances | 9K | 13K |

# D   Hyper-parameter Details and Training

The decoder used for all tasks shares the same configuration: 9 layers, 512 attention dimensions, 2048 feed-forward dimensions, and 8 attention heads. The encoder for all tasks also uses this configuration, except that the number of layers is increased to 12. For MTLA and MLA, $r$ in Eq. 8 is set to 256 and $d_h^R$ is set to 32. In MTLA, the linear layers in Eq. 13 and Eq. 16 map the 256-dimensional input to a 64-dimensional space. The temporal compression rate $s$ is set to 2 by default unless otherwise specified. Standard RoPE [39] is used in MHA to obtain positional information. Decoupled RoPE is used in MLA, which is also employed in the proposed MTLA along with the temporal compression described in Section 4.3.

For the ST task, following the Fairseq example, a Transformer encoder (including convolutional layers for 4× downsampling) is used and initialised with ASR task weights. For the text summarisation task, a standard Transformer encoder is used. For the ASR task, a Transformer encoder with convolutional layers for 2× downsampling is employed. In addition to the cross-entropy loss, a connectionist temporal classification (CTC) [18] auxiliary loss is computed with weight 1. For the SLU task, a Conformer [20] encoder is used; beyond the configuration (e.g. 512 attention dimension) used in

Transformer, its depthwise convolutional layer has a kernel size of 31. Before entering the Conformer, a convolutional layer with 2× downsampling is applied.

For the ST task, training follows the Fairseq example, using a learning rate of 2e-3, 10,000 warm-up steps, and a maximum of 100,000 update steps. Each batch corresponds to 320,000 frames of Fbank features, which is approximately 53 minutes of speech. The MHA, MLA, and MTLA models all have 78M parameters. The GQA and MQA models constructed in Section 6.3 have 74M parameters. During inference, each batch corresponds to 50,000 frames of Fbank features, and the beam size is set to 50. For the text summarisation task, training uses a learning rate of 2e-4, 15,000 warm-up steps, and a maximum of 60,000 update steps. Each batch corresponds to 40000 tokens. The MHA, MLA, and MTLA models all have 79M parameters. During inference, each batch corresponds to 60,000 tokens, and the beam size is set to 10. For the ASR task, training uses a learning rate of 2e-4, 15,000 warm-up steps, and a maximum of 10,000 update steps. Each batch corresponds to approximately 16 minutes of speech. The MHA, MLA, and MTLA models all have 67M trainable parameters. During inference, each batch corresponds to 20 minutes of speech, and the beam size is set to 20. For the SLU task, training uses a learning rate of 2e-4, 50,000 warm-up steps, and a maximum of 30,000 update steps. Each batch corresponds to 18,000 frames of Fbank features. The MHA, MLA, and MTLA models all have 103M parameters. During inference, each batch corresponds to 130,000 frames of Fbank features, and the beam size is set to 10.

Model training was performed on a single NVidia RTX 6000 Ada GPU with 48GB of memory. For the ST task, each epoch took about 13 minutes. For the text summarisation, each epoch took about 20 minutes. For the ASR task, each epoch took about 50 minutes. For the SLU task, each epoch took about 15 minutes.

# E  Results on Long Range Arena

Table 8: Experimental results on Long-Range Arena benchmark [41]. The published results other than MTLA are taken from [41].

| Model | Listops($\uparrow$) | Text($\uparrow$) | Retrieval($\uparrow$) | Image($\uparrow$) | Pathfinder($\uparrow$) | Avg($\uparrow$) |
|---|---|---|---|---|---|---|
| Transformer | 36.37 | 64.27 | 57.46 | 42.44 | 71.40 | 54.39 |
| Local Attention | 15.82 | 52.98 | 53.39 | 41.46 | 66.63 | 46.06 |
| Sparse Transformers | 17.07 | 63.58 | 59.59 | 44.24 | 71.71 | 51.24 |
| Longformer | 35.63 | 62.85 | 56.89 | 42.22 | 69.71 | 53.46 |
| Linformer | 35.70 | 53.94 | 52.27 | 38.56 | 76.34 | 51.36 |
| Reformer | 37.27 | 56.10 | 53.40 | 38.07 | 68.50 | 50.67 |
| Sinkhorn Transformers | 33.67 | 61.20 | 53.83 | 41.23 | 67.45 | 51.39 |
| Synthesizer | 36.99 | 61.68 | 54.67 | 41.61 | 69.45 | 52.88 |
| BigBird | 36.05 | 64.02 | 59.29 | 40.83 | 74.87 | 55.01 |
| Linear Transformer | 16.13 | 65.90 | 53.09 | 42.34 | 75.30 | 50.55 |
| Performer | 18.01 | 65.40 | 53.82 | 42.77 | 77.05 | 51.41 |
| Proposed MTLA | 40.47 | 66.99 | 59.88 | 48.10 | 67.55 | 56.60 |

This section further evaluates MTLA on the Long-Range Arena (LRA) benchmark [41]. The LRA benchmark was not included in the main text, as it is not primarily designed for decoder-only architectures and is therefore more suitable for evaluating encoder self-attention mechanisms. As shown in Table 8, MTLA achieves strong performance on the LRA benchmark compared to other attention mechanisms. These results complement the findings presented in the main text, demonstrating that MTLA consistently performs well across diverse modalities, including text, speech, and vision. The consistent performance across tasks indicates that MTLA effectively captures long-range dependencies while maintaining computational efficiency. It is worth noting that state space models (SSMs) are known to outperform Transformer-based attention mechanisms on the LRA benchmark due to their formulation as long convolutions with time-decay dynamics, which are particularly advantageous for tasks with strong positional dependencies. Nevertheless, we include Table 8 because LRA remains a challenging and well-established benchmark for attention mechanisms, providing a valuable test of MTLA's capability under difficult long-context conditions.

## F    Results on Machine Translation

Table 9: Machine translation BLEU ($\uparrow$) results on WMT14 [6] English-German translation

| Model on WMT14 En-De | BLEU |
|---|---|
| MLA | 25.63 |
| Proposed MTLA | 25.57 |

While the focus of this paper is on long-sequence tasks, text-based translation generally involves much shorter sequences than speech translation. Nevertheless, this section presents MTLA results on the machine translation task using WMT14 English–German data. As shown in Table 9, MTLA achieves competitive performance compared to MLA, demonstrating that context compression does not degrade performance on this task.

## G    Assets and licenses

The following licenses apply to the datasets used in this paper:

- CC-BY-NC-ND-4.0: `https://spdx.org/licenses/CC-BY-NC-ND-4.0` applies to MuST-C data.
- CC-BY-SA-4.0: `https://spdx.org/licenses/CC-BY-SA-4.0` applies to XSum data.
- CC BY 4.0: `https://spdx.org/licenses/CC-BY-4.0` applies to AMI data.
- CC BY-NC 4.0: `https://spdx.org/licenses/CC-BY-NC-4.0` applies to SLURP data.

The following license applies to the code and Python package used in this paper:

- Apache-2.0: applies to Fairseq (`https://github.com/facebookresearch/fairseq/blob/main/LICENSE`).

