# OpenReview forum: "Multi-head Temporal Latent Attention"
_NeurIPS.cc/2025/Conference — NeurIPS 2025 poster_

### Official Review · Reviewer_hHHE · 2025-06-28

**Clarity:** 3
**Significance:** 3
**Originality:** 3
**Rating:** 5
**Confidence:** 5

**Summary:**

This paper proposes MTLA, a novel method that compresses the KV cache along the temporal dimension. This method is orthogonal to other compression methods along the hidden dimension. Experiments across multiple tasks show that MTLA greatly accelerates inference and reduces GPU memory usage at inference without sacrificing accuracy.

**Questions:**

* If my understanding to the algorithm is correct, Figure 2 (c) is incorrect. Concretely, the first element of each row (instead of each column) should be q1, q2, q3 and q4, respectively.

**Ethical Concerns:**

["NO or VERY MINOR ethics concerns only"]

**Final Justification:**

Overall, this is an inspiring paper that temporally compresses the KV cache. Although the computational resource limits this paper,  I think the empirical study on a relative small dataset and model size has shown promising results. Therefore, I lean to accept this paper.

**Limitations:**

See Weaknesses.

**Quality:**

3

**Strengths And Weaknesses:**

Strengths:
* This is a very novel method that successfully compress KV cache along the temporal dimension.
* Experiments also show promising results.
* The paper is well written and easy to follow.

Weakness:
* Although the proposed method could be a promising acceleration solution to LLMs, the authors only conduct experiments on relative small settings (d=512). It would be great if authors could add experiments on LM tasks. Even a 1~7B LM model could make this work more siginificant.

---

> ### Author Rebuttal · Authors · 2025-07-30
>
> We appreciate the encouraging comments in the summary and strengths! It's encouraging when the motivation and strengths of our work are understood. Below, we provide a point-by-point response to the reviewer's comments.
>
> - Weakness 1:
>     > Although the proposed method could be a promising acceleration solution to LLMs, the authors only conduct experiments on relative small settings (d=512). It would be great if authors could add experiments on LM tasks. Even a 1~7B LM model could make this work more siginificant.
>     - Thanks for the valuable suggestion. We agree that it would be very useful to pretrain an LLM based on MTLA. However, as we mentioned in the Limitations section, we are constrained by computational resources and currently cannot afford the cost of pretraining such a model. Even at the scale of 1-7B parameters, ensuring a fair comparison with other 1-7B parameter LLMs still requires large-scale training data and extensive pretraining effort. This is a challenge we will need to address in future, but we believe the experimental validation we currently provide should be acceptable within the scope of an academic paper. We do plan to actively promote the widespread adoption of MTLA and KV cache temporal compression, so scaling to LLMs is definitely an important future direction.
>
> - Question 1:
>     > If my understanding to the algorithm is correct, Figure 2 \(c) is incorrect. Concretely, the first element of each row (instead of each column) should be q1, q2, q3 and q4, respectively.
>     - Thanks for pointing this out. We will fix this typo in the final version of the paper.

---

> > ### Comment · Reviewer_hHHE · 2025-08-05
> > **Response to Rebuttal**
> >
> > I've read the rebuttal, and I'm a little disappointed to hear that computational resources limit this paper. I'll keep my original scores.

---

### Official Review · Reviewer_6bRo · 2025-07-01

**Clarity:** 3
**Significance:** 3
**Originality:** 2
**Rating:** 4
**Confidence:** 3

**Summary:**

This paper optimizes KV cache for attention-based inference by introducing Multi-head Temporal Latent Attention, which compresses KV cache along the temporal dimension via dynamic merging of adjacent tokens and adjusts training-inference methods to reduce overhead.

**Questions:**

1.	What is the practical or theoretical significance of merging latent embeddings of tokens temporally?
2.	Does the proposed method alter the model training process, and how do training time and parameter space perform?
3.	Could you provide parameter implementations under different s values to further clarify the real effects of temporal merging and offer deeper insights?

**Ethical Concerns:**

["NO or VERY MINOR ethics concerns only"]

**Final Justification:**

I think this work presents a clear idea and validates it experimentally, but the analysis lacks depth, so I'm inclined to keep the rating unchanged.

**Limitations:**

Yes

**Quality:**

3

**Strengths And Weaknesses:**

Strengths:
1.	The paper presents a clear and well-structured introduction to the method, making it easy to follow the design rationale and model workflow.
2.	It provides a logical review of related work, clearly highlighting the distinctions between the proposed approach and existing methods.
3.	Experiments on speech tasks demonstrate significant inference speedups while maintaining accuracy, showcasing practical efficiency gains.
Weaknesses:
1.	Although temporal merging shows effectiveness in experiments, the paper lacks in-depth analysis of its principles and validity, requiring more theoretical insights.
2.	The tested scenarios focus on long-sequence speech tasks, and the generalizability of the approach across broader applications remains unvalidated.
3.	The compression ratio s is a hyperparameter that requires manual tuning, but the paper does not offer guidance on optimal setting strategies or insights into its impact.

---

> ### Author Rebuttal · Authors · 2025-07-30
>
> We appreciate the encouraging comments in the summary and strengths! Below, we provide a point-by-point response to the reviewer's comments.
>
> - Weakness 1:
>     > 1. Although temporal merging shows effectiveness in experiments, the paper lacks in-depth analysis of its principles and validity, requiring more theoretical insights.
>     - In the original manuscript, specifically in lines 39 to 42 of the Introduction, we provided our analysis and insights. In particular, previous works such as MLA focus on compressing the feature dimension of the KV cache, and approaches like GQA or MQA reduce the number of attention heads. While these approaches have demonstrated effectiveness, they may face inherent constraints. For example, the number of attention heads cannot be reduced below one. In contrast, as modern models produce increasingly long outputs, the length of the KV cache also becomes significantly larger. This makes compression along the temporal dimension a promising direction. This idea is also theoretically grounded, considering that recurrent neural networks can be seen as a special case of attention where the KV cache always has a length of one. Therefore, our proposed temporal compression technique is both reasonable and well aligned with emerging practical needs. We also revisited the potential of this direction in the experimental section, specifically in Section 6.3, lines 298 to 302.
>
> - Weakness 2:
>     > 2. The tested scenarios focus on long-sequence speech tasks, and the generalizability of the approach across broader applications remains unvalidated.
>     - In the original manuscript, we conducted a series of experiments on speech tasks, as well as a text summarisation task, because both speech and documents involve long-sequence scenarios.
>     - To further address the reviewer’s concerns, we have now conducted additional validation on more tasks. We additionally conducted evaluations on the Long-Range Arena (LRA) benchmark (Tay et al., 2021).LRA is a widely used benchmark for testing efficient Transformer models in long-context scenarios. We did not include this benchmark in the original version of the paper because LRA is not primarily designed for decoder-only architectures and may be more suitable for encoder self-attention. Below, we present the results of MTLA on the LRA benchmark (with other results taken from the Tay et al., 2021 paper):
>         | Model| Listops（`↑`）| Text（`↑`） |Retrieval（`↑`）| Image（`↑`） |Pathfinder（`↑`） | Avg（`↑`）|
>         | :----------- | :-------: | :-------: |:-------: |:-------: |:-------: | -------:|
>         |Transformer| 36.37| 64.27| 57.46| 42.44 |71.40|54.39|
>         |Local Attention| 15.82| 52.98| 53.39| 41.46| 66.63|  46.06|
>         |Sparse Transformers| 17.07 |63.58| 59.59| 44.24| 71.71 | 51.24|
>        | Longformer |35.63| 62.85 |56.89 |42.22| 69.71| 53.46|
>       | Linformer| 35.70 |53.94| 52.27| 38.56| 76.34| 51.36|
>        |Reformer| 37.27 |56.10| 53.40| 38.07| 68.50 |50.67|
>        |Sinkhorn Transformers| 33.67 |61.20 |53.83 |41.23 |67.45 | 51.39|
>        |Synthesizer |36.99| 61.68 |54.67| 41.61 |69.45| 52.88|
>        |BigBird| 36.05 |64.02 |59.29| 40.83| 74.87| 55.01|
>        |Linear Transformer| 16.13| 65.90| 53.09| 42.34| 75.30| 50.55|
>        |Performer |18.01| 65.40| 53.82| 42.77| 77.05|51.41|
>        |**MTLA**|40.47|66.99|59.88|48.1|67.55|**56.60**|
>
>       "ListOps" is a 10-way classification task with a sequence length of 2K. "Text" is a binary classification task with sequences up to 2,048 tokens. "Retrieval" is also a binary classification task, requiring processing of 8K tokens during evaluation. "Image" is a 10-way classification task with a sequence length of 1,024. "Pathfinder" is a binary image classification task with sequences of length 1,024. For more detail about LRA refer to Tay at al. (2021).
>
>     - The results above show that MTLA achieves competitive performance on the LRA benchmark compared to other attention mechanisms. Combined with the experiments in the original paper, MTLA has demonstrated consistently strong performance across a range of tasks involving text, speech, and vision.
>
>     Tay et al. 2021. "Long range arena: A benchmark for efficient transformers." In Proc. ICLR.
>
> - Weakness 3:
>     > 3. The compression ratio s is a hyperparameter that requires manual tuning, but the paper does not offer guidance on optimal setting strategies or insights into its impact.
>     - This paper has already recommended setting the default value of compression ratio “s” to 2, since in this case the KV cache size is comparable to that of MQA. We have mentioned this point in several places in the original manuscript (lines 200, 227, 259, 290, and 293). Increasing the value of s results in a higher degree of KV Cache compression, but it may lead to some decrease in model accuracy, as shown in Table 5 of the original manuscript. There is no single optimal value of "s" that applies to all situations. The best choice depends on the specific trade-off between performance and efficiency required by the application. For this reason, we recommend setting the default value to 2.
>
> - Question 1:
>     > What is the practical or theoretical significance of merging latent embeddings of tokens temporally?
>     - Our work focuses on compressing the key/value (KV) cache of the Transformer self-attention mechanism along the temporal dimension during inference. In autoregressive generation with Transformers, the KV cache grows linearly with the sequence length and becomes a bottleneck for inference. Each decoding step requires loading the cached attention keys and values to avoid re-encoding the history. This repeated memory access has become a limiting factor, affecting inference speed and restricting both decoding batch size and sequence length. Therefore, compressing the KV cache can effectively reduce this memory bandwidth overhead, which is crucial for efficient deployment. As we discussed in our response to Weakness 1, since modern models generate increasingly longer sequences, compressing the KV cache along the temporal dimension holds significant potential. This direction of research can substantially improve Transformer inference efficiency, and our experimental results support this claim.
>
> - Question 2:
>     > Does the proposed method alter the model training process, and how do training time and parameter space perform?
>     - As described in Section 4.2 of the original manuscript, MTLA is fully parallelisable during training and shares the same computational and memory complexity as standard attention. Therefore, our proposed method does not alter the model training process.
>
> - Question 3:
>     > Could you provide parameter implementations under different s values to further clarify the real effects of temporal merging and offer deeper insights?
>     - As shown in Figures 1 and 2 of the original manuscript, "s" is used to control the degree of temporal compression of the KV Cache. Specifically, if s = 2, every two adjacent KV Cache vectors are merged into one vector, and the total length of the KV Cache becomes half of the original; if s = 3, every three adjacent KV Cache vectors are merged into one vector, and the total length of the KV Cache becomes one-third of the original; and so on. Therefore, by adjusting "s", the compression level of the KV Cache can be well controlled.
>
> We hope that our clarifications have effectively addressed your concerns. We hope you will kindly consider the value and contribution of our work in your final evaluation, taking into account our responses to your review.

---

> > ### Comment · Reviewer_6bRo · 2025-08-07
> >
> > Thank you for your response and the additional experiments on more tasks, which are valuable supplements. While the motivation for temporal merging has been clearly presented, I still suggest that the authors conduct a more in-depth analysis of why this mechanism is effective. Overall, I will maintain my original score.

---

> > > ### Author Response · Authors · 2025-08-07
> > > **Analysis of the MTLA mechanism effectiveness**
> > >
> > > We are pleased to have addressed some of the reviewer’s concerns and to have clearly presented the motivation behind our temporal compression. Below, we would like to further provide some mathematical analysis that shows how  the MTLA mechanism operates and why it is able to be is effective.
> > >
> > > First, the essence of attention is a weighted summation. For example, suppose the sequence of attention values is $(V_1, V_2, V_3, V_4)$. In the standard attention mechanism, after multiplying the query and keys, we obtain the weights corresponding to the values, denoted as $( a_1, a_2, a_3, a_4)$. The output is then calculated as:
> > >
> > > $a_1 \cdot V_1 + a_2 \cdot V_2 + a_3 \cdot V_3 + a_4 \cdot V_4$
> > >
> > > In MTLA, during temporal compression, adjacent KV caches are also merged in a weighted manner. For example, the sequence of attention values becomes:
> > >
> > > $\hat{V}_1 = w_1 \cdot V_1 + w_2 \cdot V_2,\quad \hat{V}_2 = w_3 \cdot V_3 + w_4 \cdot V_4$
> > >
> > > where the weights $( w_1, \dots, w_4 )$ are dynamically generated by a hypernetwork. Therefore, we only need to cache $\hat{V}_1$ and $\hat{V}_2$, which improves inference efficiency.
> > >
> > > Similarly, MTLA, like standard attention, multiplies the query with the keys to obtain the weights $( a_1, a_2 )$, and produces the output:
> > >
> > > $a_1 \cdot \hat{V}_1 + a_2 \cdot \hat{V}_2$
> > >
> > > This can be rewritten as:
> > >
> > > $(a_1 \cdot w_1) \cdot V_1 + (a_1 \cdot w_2) \cdot V_2 + (a_2 \cdot w_3) \cdot V_3 + (a_2 \cdot w_4) \cdot V_4$
> > >
> > > Letting $\hat{a}_1 = a_1 \cdot w_1$, $\hat{a}_2 = a_1 \cdot w_2$, $\hat{a}_3 = a_2 \cdot w_3$, and $\hat{a}_4 = a_2 \cdot w_4$, the expression becomes:
> > >
> > > $\hat{a}_1 \cdot V_1 + \hat{a}_2 \cdot V_2 + \hat{a}_3 \cdot V_3 + \hat{a}_4 \cdot V_4$
> > >
> > > Therefore, in essence, MTLA remains very similar to the standard attention mechanism as it only changes the weight distribution, allowing effective temporal compression of the KV cache. Considering the inherent flexibility of attention mechanisms, their weight distributions are both adaptable and robust. For example, each head in a multi-head attention mechanism typically assigns very different weights. Therefore, it is reasonable that MTLA, which dynamically adjusts weight distribution through a hypernetwork, is effective.
> > >
> > > We thank the reviewer for their valuable suggestions. We will include the above analysis in an appendix to the final version of the paper as we feel it clarifies how MTLA works and why it is effective. We would still kindly hope that the reviewer takes our additional efforts and results into consideration when making their final evaluation.

---

### Official Review · Reviewer_yMqX · 2025-07-02

**Clarity:** 3
**Significance:** 2
**Originality:** 3
**Rating:** 4
**Confidence:** 4

**Summary:**

The paper proposes a method to compress the self-attention KV-cache along the temporal dimension, extending the recent Multi-head Latent Attention (MLA) technique to further reduce compute and memory overhead.

Since the proposed approach modifies the KV cache temporal sequence during attention, special care must be taken during training to ensure that it is consistent with the partial decimation in time that must occur during token-by-token sampling during inference.

Experiments on small scale speech and text tasks demonstrate significant memory/speed improvements without hurting performance compared to a vanilla attention baseline and other commonly used attention optimizations.

**Questions:**

See specific suggestions for improvements listed in Weakness above.

Minor issues/questions not raised above:

 - I found Fig 1 to be a bit confusing.  Shouldn't the "Multi-head Queries" also be derived from the "Attention Input"?  Also, it would be even more clear if the different components were explicitly connected to the notation introduced in Sec 3.

- Sec 4.3, line 184:  Did you mean "Eq. 9" instead of "Eq. 8"?

- Sec 4.3, line 201: $9d_hl/(2s)$ would be $2.25 d_hl$, not $2.5 d_hl$ if $s=2$.

- Table 5 repeats most of Table 1.  These can probably be consolidated into a single table.

- The final two sentences of the first paragraph of Sec. 5.2 (lines 221-224) are near exact repeats of the preceding two sentences.

- End of Sec 6.3: Why only comment on the significance of one result?  Are the performance improvements of "Proposed MTLA w/$s=4$" over the other baselines not significant?

**Ethical Concerns:**

["NO or VERY MINOR ethics concerns only"]

**Final Justification:**

The author's additional experimental validation help to strengthen the minor criticisms of the work.  But the main weakness remains: I expect that most readers will wonder (as did the reviewers) how well the proposed method applies at LLM scale.

In my opinion the simplicity and demonstrated effectiveness of the method at small scale are intriguing enough to overcome this weakness.  But I also understand if others would disagree, hence my borderline recommendation.

**Limitations:**

Yes.  The authors explicitly acknowledge the limitations of the small scale experiments due to limited compute resources available to them.  This is totally reasonable.  However, as described above, it feels like an oversight to focus exclusively on long sequence tasks in the experimental evaluation with no other justification.  Some experimental comparisons on other text tasks (at small scale) would give a more rounded perspective on how broadly applicable MLTA is.

**Paper Formatting Concerns:**

No issues.

**Quality:**

3

**Strengths And Weaknesses:**

Strengths

- Straightforward, well motivated approach to reduce self-attention compute/memory requirements, extending recent work on latent attention.
- Experimental validation of decoder-only transformer across several different tasks.
- Good results - large inference speedups over latent attention baseline.

Weaknesses

- The main weakness is that the experimental evaluation is on on very small scale models of around 100M parameters, more than an order of magnitude smaller than the smallest typical LLM size (~2-3B parameters).  The author's justify this choice due to limited access to compute, which is totally understandable, but this might limit interest in the work.

- Curious choice of evaluation tasks, focusing primarily on speech tasks.  The proposed technique seems quite generally applicable to any sequence modeling task.  Is there a reason for not including incorporating other NLP tasks, e.g., machine translation (which is supported in fairseq) in the experiments?  It is reasonable to assume that the proposed technique better suited to speech tasks because sequence lengths are longer.  And perhaps speech signals are more naturally compressible in time than tokenized text, making them better suited to the proposed temporal compression technique?  But it would nevertheless be valuable to include some results on some other NLP tasks, even if sequences are shorter, to validate that compressing context does not hurt performance on such tasks.

   - Similarly $s > 2$ is only evaluated on the speech translation task.  It would be valuable to see how MTLA performs on other tasks with more compression., especially XSum since it is the only text-only task considered in the paper.

- Some design choices not thoroughly justified/evaluated, especially the construction of the hypernetwork that compute the merging weights $w_i$ in Eq. 13.  Are these details important?  Does performance suffer if $w_i = 1/s$ (simple averaging) or some other constant?  If nothing else, some discussion motivating this choice  seems warranted.  Even better would be a quick ablation experiment comparing the proposed technique to a simple baseline.

---

> ### Author Rebuttal · Authors · 2025-07-30
>
> Thanks for your very detailed review and insightful comments. It's encouraging when the motivation and strengths of our work are clearly understood. We provide a point-by-point response to your review below.
>
> - Weakness 1:
>     > The main weakness is that the experimental evaluation is on on very small scale models of around 100M parameters, more than an order of magnitude smaller than the smallest typical LLM size (~2-3B parameters). The author's justify this choice due to limited access to compute, which is totally understandable, but this might limit interest in the work.
>      - Thanks for pointing this out. We agree that it would be interesting if we can afford to pretrain an LLM based on MTLA. However, as we mentioned in the Limitations section, due to limited computational resources, we are unable to afford LLM pretraining, which is why we did not explicitly link MTLA to LLMs in the main text of the original manuscript. This is a challenge we will need to address in the future, but we believe the experimental validation we currently provide should be acceptable within the scope of an academic paper.
> We do plan to actively promote the widespread adoption of MTLA and KV cache temporal compression, so investigating MTLA with LLMs is a clear future direction.
>
>
>
>
> - Weakness 2:
>     > Curious choice of evaluation tasks, focusing primarily on speech tasks. The proposed technique seems quite generally applicable to any sequence modeling task. Is there a reason for not including incorporating other NLP tasks, e.g., machine translation (which is supported in fairseq) in the experiments? It is reasonable to assume that the proposed technique better suited to speech tasks because sequence lengths are longer. And perhaps speech signals are more naturally compressible in time than tokenized text, making them better suited to the proposed temporal compression technique? But it would nevertheless be valuable to include some results on some other NLP tasks, even if sequences are shorter, to validate that compressing context does not hurt performance on such tasks.
>     - This paper has been validated on various speech tasks and text summarisation tasks. As explained in Section 6 of the original manuscript, the focus of this work is on long-sequence tasks, and both speech and document inputs naturally involve long sequences. In contrast, text-based translation tasks generally involve much shorter sequences than speech, and since they are all translation tasks, including them would introduce some unnecessary redundancy. Therefore, we did not include them in the original manuscript. Following the reviewer’s suggestion, we conducted additional experiments on the WMT14 English-German translation task using Fairseq. These results are shown below.
>         | Model on WMT14 En-De | BLEU |
>         | :-----------  | -------:|
>         |MLA|25.63|
>         |MTLA|25.57|
>
>         The results show that MTLA achieves competitive performance with MLA in machine translation. Furthermore, to further address the reviewer’s concerns, we have now conducted additional validation on more tasks. We additionally conducted evaluations on the Long-Range Arena (LRA) benchmark (Tay et al., 2021).LRA is a widely used benchmark for testing efficient Transformer models in long-context scenarios. We did not include this benchmark in the original version of the paper because LRA is not primarily designed for decoder-only architectures and may be more suitable for encoder self-attention.
>     - Below, we present the results of MTLA on the LRA benchmark (with other results taken from the Tay et al., 2021 paper):
>         | Model| Listops（`↑`）| Text（`↑`） |Retrieval（`↑`）| Image（`↑`） |Pathfinder（`↑`） | Avg（`↑`）|
>         | :----------- | :-------: | :-------: |:-------: |:-------: |:-------: | :-------:|
>         |Transformer| 36.37| 64.27| 57.46| 42.44 |71.40|54.39|
>         |Local Attention| 15.82| 52.98| 53.39| 41.46| 66.63|  46.06|
>         |Sparse Transformers| 17.07 |63.58| 59.59| 44.24| 71.71 | 51.24|
>        | Longformer |35.63| 62.85 |56.89 |42.22| 69.71| 53.46|
>       | Linformer| 35.70 |53.94| 52.27| 38.56| 76.34| 51.36|
>        |Reformer| 37.27 |56.10| 53.40| 38.07| 68.50 |50.67|
>        |Sinkhorn Transformers| 33.67 |61.20 |53.83 |41.23 |67.45 | 51.39|
>        |Synthesizer |36.99| 61.68 |54.67| 41.61 |69.45| 52.88|
>        |BigBird| 36.05 |64.02 |59.29| 40.83| 74.87| 55.01|
>        |Linear Transformer| 16.13| 65.90| 53.09| 42.34| 75.30| 50.55|
>        |Performer |18.01| 65.40| 53.82| 42.77| 77.05|51.41|
>        |**MTLA**|40.47|66.99|59.88|48.1|67.55|**56.60**|
>
>
>       "ListOps" is a 10-way classification task with a sequence length of 2K. "Text" is a binary classification task with sequences up to 2,048 tokens. "Retrieval" is also a binary classification task, requiring processing of 8K tokens during evaluation. "Image" is a 10-way classification task with a sequence length of 1,024. "Pathfinder" is a binary image classification task with sequences of length 1,024. For more detail about LRA refer to Tay at al. (2021).
>
>     - The results above show that MTLA achieves competitive performance on the LRA benchmark compared to other attention mechanisms. Combined with the experiments in the original paper, MTLA has demonstrated consistently strong performance across a range of tasks involving text, speech, and vision.
>
>     Tay et al. 2021. "Long range arena: A benchmark for efficient transformers." In Proc. ICLR.
>
> - Weakness 3:
>     > Similarly s>2 is only evaluated on the speech translation task. It would be valuable to see how MTLA performs on other tasks with more compression., especially XSum since it is the only text-only task considered in the paper.
>     - In the original manuscript, we set s=2 as the default value for MTLA, as it makes the KV cache size comparable to MQA. Following the reviewer’s suggestion, we include below the results on XSum with s > 2:
>         | Model on XSum | R1 | R2|RL|
>         | :-----------  | :-------:| :-------:| :-------:|
>         |MHA|28.83|9.67|23.33|
>         |MTLA|29.14|9.79|23.60|
>         |MTLA with s=3|28.80|9.55|23.33|
>         |MTLA with s=4|28.49|9.49|23.10|
>
>     The results above show that with a higher compression rate, i.e., a larger value of "s", MTLA can still achieve competitive performance. If the target task involves longer and more redundant information, "s" could theoretically be set to a larger value. However, in this paper, we prefer to recommend s = 2 as the default setting.
>
> - Weakness 4:
>     > Some design choices not thoroughly justified/evaluated, especially the construction of the hypernetwork that compute the merging weights w_i
>  in Eq. 13. Are these details important? Does performance suffer if w_i = 1/s (simple averaging) or some other constant? If nothing else, some discussion motivating this choice seems warranted. Even better would be a quick ablation experiment comparing the proposed technique to a simple baseline.
>      - Using a hypernetwork to generate the weights w_i allows for dynamically merging temporally adjacent KV cache vectors. Following the reviewer’s suggestion, we conducted an additional ablation study comparing our approach to the suggested baseline of w_i = 1/s. The results are shown below:
>          | ST Model  | Quality (BLEU) |
>         | :----------- |  :-------:|
>         |MTLA with w_i = 1/s|22.90|
>         |MTLA|23.28|
>
>         The results show that using a hypernetwork to dynamically generate the weights yields better performance, which is also intuitively reasonable.
>
> - Question 1:
>     > I found Fig 1 to be a bit confusing. Shouldn't the "Multi-head Queries" also be derived from the "Attention Input"? Also, it would be even more clear if the different components were explicitly connected to the notation introduced in Sec 3.
>     - We appreciate such detailed suggestions. The multi-head query indeed originates from the attention input. We could add additional arrows from the attention input to the multi-head query to make this clearer, although doing so would slightly increase the visual complexity of the figure. Thank you for the valuable feedback. We will make some adjustments in the final version to improve clarity.
>
> - Question 2:
>     > Sec 4.3, line 184: Did you mean "Eq. 9" instead of "Eq. 8"?
>     > Sec 4.3, line 201: 9d_hl/(2s) would be 2.25 d_hl, not 2.5 d_hl if s=2
>     - We appreciate these detailed comments and will address these typos in the final version of the paper.
>
> - Question 3:
>     > Table 5 repeats most of Table 1. These can probably be consolidated into a single table.
>     - The reason we did this is to make reading easier, so that readers do not need to flip back and forth between pages for comparison. We will consider your suggestion for the final version of the paper.
>
> - Question 4:
>     > The final two sentences of the first paragraph of Sec. 5.2 (lines 221-224) are near exact repeats of the preceding two sentences.
>     - Thanks for pointing this out. We will make sure to fix this issue in the final version of the paper.
>
> - Question 5:
>     > End of Sec 6.3: Why only comment on the significance of one result? Are the performance improvements of "Proposed MTLA w/ s=4" over the other baselines not significant?
>     - This is because in that paragraph (lines 298–302), we focus on the comparison between MTLA and MQA, as MTLA is comparable to MQA in terms of KV cache size when s=2. The reason we conducted statistical testing was to show that even with s=4, MTLA still achieves better translation quality than MQA.
>
> We hope that our clarifications have effectively addressed your concerns. We hope you will kindly consider the value and contribution of our work, along with the additional information provided here, in your final review scores.

---

> > ### Comment · Reviewer_yMqX · 2025-08-04
> >
> > I'm grateful for the authors' efforts in performing additional experiments.
> >
> > > This paper has been validated on various speech tasks and text summarisation tasks. As explained in Section 6 of the original manuscript, the focus of this work is on long-sequence tasks, and both speech and document inputs naturally involve long sequences.
> >
> > Indeed, and as I suggested in my review the goal of including other benchmarks would be to validate that the proposed technique does not hurt performance on shorter sequence tasks.  The new MT (and LRA) results are useful indication that this might not be an issue, and I encourage the authors to include them in the final manuscript somewhere (e.g., in an appendix if necessary).
> >
> > Following the other reviewers it would be ideal to cite and compare MLTA to related work on Dynamic Memory Compression  and Activation Beacons.
> >
> > Overall I stand by my original review.  I think this work has merit, even if the evaluation is not as exhaustive nor as large scale as might be ideal.  But the promising initial results in the current paper make experimenting at larger scale the obvious step for followup work.

---

> > > ### Author Response · Authors · 2025-08-05
> > > **Following Response to Reviewer yMqX**
> > >
> > > Thank you for getting back to us. We will make sure to include these additional results in the final version of the paper since the additional experiments suggested by the reviewer were very helpful in strengthening the paper. In the Related Work section of the original manuscript we discussed some approaches related to dynamic token pruning and, in the final version of the paper, we will of course also cite and discuss Dynamic Memory Compression and Activation Beacons.
> > >
> > > As a new attention mechanism, MTLA offers certain advantages over these methods. For example, token merging often requires changes to inference or training, while MTLA is a drop-in attention mechanism that requires no changes beyond the attention module, which is beneficial for deployment. MTLA can be easily applied across modalities such as speech, text, and vision, whereas some compression methods are less suitable for continuous inputs like speech. Moreover, MTLA does not reduce the number of output tokens but instead shortens KV caches, making it potentially complementary to token merging.
> > >
> > > Overall, we thank the reviewer for helping us improve our paper. In particular, as the reviewer pointed out, we found that MTLA may not hurt performance on shorter sequences, which is a valuable insight. We believe our responses and additional experiments have addressed the reviewer’s concerns, except for large-scale LLM pretraining, which we are currently unable to computationally afford and this paper is not LLM-focused. However, we will definitely explore this direction in the future. We would still kindly hope that the reviewer takes our additional efforts and results into consideration when making their final evaluation.

---

### Official Review · Reviewer_xgRJ · 2025-07-03

**Clarity:** 3
**Significance:** 3
**Originality:** 3
**Rating:** 5
**Confidence:** 4

**Summary:**

Current LLM architectures face severe memory constraints during inference. To address this challenge, the authors introduce a novel technique that compresses the KV-cache along the temporal dimension using a learnable approach. They report over 3× speedups and substantial VRAM savings during inference compared to vanilla multi-head attention.

**Questions:**

1. Can this approach be combined with other head reduction techniques such as GQA?
2. Have you experimented with FP16/BF16 precision for training and inference?
3. While resource constraints prevent testing on large-scale LLMs, what evidence suggests this approach will scale effectively to larger models? Could you provide scaling law experiments, even if limited to smaller model sizes?

**Ethical Concerns:**

["NO or VERY MINOR ethics concerns only"]

**Final Justification:**

This work presents MTLA, an extension to Multi-head Latent Attention where compression is also applied on the sequence length dimension, effectively reducing the size of the KV-Cache and improving inference speed.

The results look promising, and the authors have addressed all the weaknesses.
We reckon that the main weakness of the paper, the small scale of the experiments, cannot be easily resolved due to the resource limitations of the authors. Nevertheless, the authors successfully test their method on a variety of tasks, and are planning on releasing the implementation to the public. We believe that the strengths outweigh the weaknesses, and that MTLA opens an exciting area of future research on efficient attention architectures.

**Limitations:**

Yes

**Quality:**

3

**Strengths And Weaknesses:**

# Strengths
- The authors present Multi-head Temporal Latent Attention (MTLA), a novel self-attention variant that pioneers learnable compression of the KV-cache along the temporal axis while preserving the key-value compression from MLA. This approach opens a promising new research direction for improving inference-time efficiency in Transformer-based models.
- The research addresses a well-motivated problem, as current LLMs face significant memory bottlenecks due to KV-cache growth. The proposed approach introduces clever technical solutions with sound theoretical explanations.
- The empirical results are compelling: the method demonstrates consistent performance across diverse tasks (speech translation, ASR, text summarization, and SLU) while matching the accuracy of previous methods. This suggests that the compression effectively removes redundancy from the KV-cache without sacrificing model quality.
# Weaknesses
- **Novelty claim**: While the authors claim this is the first work to explore KV-cache compression along the temporal axis, [Nawrot et al. (2024)](https://arxiv.org/abs/2403.09636) have already investigated temporal compression of the KV-cache. Although their approach focuses on retrofitting pre-trained LLMs rather than making fundamental architectural changes, this prior work should be acknowledged.
- **Limited comparison against baselines**: The paper does not empirically compare against established KV-cache pruning methods such as LazyLLM and SnapKV, despite mentioning them in the related work.
- **Limited comparison to other orthogonal approaches**: the KV-cache can be compressed with other methods (e.g. quantization). The authors do not compare the speed-ups obtained by MTLA to other orthogonal methods, or a combination of both.
- **GPU memory usage reporting**: The analysis of GPU memory usage lacks detail regarding measurement methodology and the breakdown between model weights, activations, and KV-cache consumption. Moreover, the average GPU memory usage is reported, whereas a more informative measure would be the peak memory usage.
- **Benchmarks might not reflect real deployment metrics**: The reported memory usage and inference time metrics heavily depend on implementation details. Since the authors acknowledge not using optimizations like FlashAttention or compiled kernels, the results may not reflect realistic deployment scenarios.
- **Lack of confidence intervals**: The experimental results lack error bars or confidence intervals, which would strengthen the statistical validity of the comparisons given the stochastic nature of neural network training.
- **Limited scale**: The core motivation addresses memory bottlenecks that primarily affect large models processing very long sequences. However, the evaluation uses small models with approximately 0.1B parameters, which may not adequately demonstrate the method's benefits at scale.
- **Lack of evaluation on common tasks**: The evaluation lacks standard LLM benchmarks (language modeling, code generation, reasoning tasks) and crucially omits long-context evaluations (100k+ tokens) where the method's advantages should be most pronounced.

---

> ### Author Rebuttal · Authors · 2025-07-30
>
> Thanks for your very detailed review and insightful comments. It's encouraging when the motivation and strengths of our work are clearly understood. We provide a point-by-point response to your review below.
>
> - Weakness 1:
>     > While the authors claim this is the first work to explore KV-cache compression along the temporal axis, Nawrot et al. (2024) have already investigated temporal compression of the KV-cache. Although their approach focuses on retrofitting pre-trained LLMs rather than making fundamental architectural changes, this prior work should be acknowledged.
>     - Thanks for the suggested reference. In the final version, we will cite and discuss Nawrot et al. (2024).
>
> - Weakness 2:
>     > The paper does not empirically compare against established KV-cache pruning methods such as LazyLLM and SnapKV, despite mentioning them in the related work.
>     > the KV-cache can be compressed with other methods (e.g. quantization). The authors do not compare the speed-ups obtained by MTLA to other orthogonal methods, or a combination of both.
>     - Thanks for the suggesions. As mentioned in the Limitations section, Transformer-based models have been extensively developed by the community over many years, and the number of related techniques precludes an exhaustive comparison. Therefore, we chose to implement and compare only the most relevant and representative KV cache compression method. On the other hand, techniques such as KV cache pruning and quantisation are orthogonal to our approach. Including all of these methods would make our experimental setup very complex and reduce clarity. Moreover, since our method is based on MLA, some of these techniques require further adaptation before they can be applied to MLA and meaningfully compared with our MTLA. We do however investigate one cache pruning method below.
>     - In response to the reviewer's suggestion, we selected a representative KV cache pruning method, SnapKV, adapted it for use with MLA, and compared it with our MTLA. The results on the same speech translation task presented in Table 1 of the original manuscript are shown below.
>         | ST Model  | Quality (BLEU) |Inference Time (s)| Inference GPU Memory (MiB) |
>         | :----------- | :-------: |:-------: | :-------:|
>         |MLA|22.97|97.0|5065|
>         |MLA with SnapKV|21.76|80.8|4222|
>         |MTLA|23.28|65.6|2835|
>
>         The results show that SnapKV can effectively improve inference efficiency, but it comes with some performance degradation. SnapKV may still be more suitable for LLM scenarios, where there is more redundant information. In contrast, MTLA, as a fundamental attention mechanism, may have broader application potential.
>
> - Weakness 3:
>     > The analysis of GPU memory usage lacks detail regarding measurement methodology and the breakdown between model weights, activations, and KV-cache consumption. Moreover, the average GPU memory usage is reported, whereas a more informative measure would be the peak memory usage.
>     - Thanks for the suggestion. We will include more detailed information in the final version of the paper.
>
> - Weakness 4:
>     > The reported memory usage and inference time metrics heavily depend on implementation details. Since the authors acknowledge not using optimizations like FlashAttention or compiled kernels, the results may not reflect realistic deployment scenarios.
>     - Thanks for pointing this out. In the Limitations section of the original manuscript we mentioned that extending FlashAttention to MTLA was planned as future work, as it requires substantial engineering effort, while this paper focuses on presenting our novel temporal compression technique. Recently, we have indeed been actively working on extending MTLA, including the integration of FlashAttention. A CUDA kernel based on MTLA has now been implemented. Below, we include some results for MTLA using our extended version of FlashAttention on the same speech translation task presented in Table 1 of the original manuscript.
>         | ST Model  | Quality (BLEU) |Inference Time (s)| Inference GPU Memory (MiB) |
>         | :----------- | :-------: |:-------: | :-------:|
>         |MHA|23.18|281.3|18646|
>         |MHA with FlashAttention|23.16|145.7|9244|
>         |MTLA|23.28|65.6|2835|
>         |MTLA with Extended FlashAttention|23.29|36.5|1259|
>
>         As the results show, using FlashAttention speeds up computation, but it does not change the experimental conclusions. We will include these results and more detail in the appendix of the final version, and we will also open-source our FlashAttention extension for MTLA.
>
> - Weakness 5:
>     > The experimental results lack error bars or confidence intervals, which would strengthen the statistical validity of the comparisons given the stochastic nature of neural network training.
>     - Due to limited computational resources, conducting multiple training runs with different random seeds would greatly increase the training cost. Therefore, instead of varying the random seed, we chose to vary the task types and validated our method across a range of different tasks, consistently reaching the same conclusion. Additionally, we conducted statistical tests (as shown in Section 6.3) to confirm that the improvements brought by our method are statistically significant. We appreciate the reviewer’s suggestion and will mention this point in the Limitations section of the final version.
>
> - Weakness 6:
>     > The core motivation addresses memory bottlenecks that primarily affect large models processing very long sequences. However, the evaluation uses small models with approximately 0.1B parameters, which may not adequately demonstrate the method's benefits at scale.
>     > The evaluation lacks standard LLM benchmarks (language modeling, code generation, reasoning tasks) and crucially omits long-context evaluations (100k+ tokens) where the method's advantages should be most pronounced.
>     - We agree that it would further strengthen the paper if we could afford to pretrain an LLM based on MTLA. However, as we mentioned in the Limitations section, due to limited computational resources, we are unable to afford LLM pretraining, which is why we did not explicitly link MTLA to LLMs in the main text of the original manuscript. This is a challenge we want to address in future, but we believe the experimental validation we currently provide is acceptable within the scope of an academic paper. We do plan to actively promote the widespread adoption of MTLA and KV cache temporal compression, so investigating MTLA with LLMs is a clear future direction.
>
> - Question 1:
>     > Can this approach be combined with other head reduction techniques such as GQA?
>     - Yes, the KV cache time compression technique we propose can certainly be combined with other attention mechanisms, including GQA.
>
> - Question 2:
>     > Have you experimented with FP16/BF16 precision for training and inference?
>     - We have tested BF16 training and found that MTLA works well with it. However, in order to ensure a fair comparison across all constructed models, we ultimately did not use BF16 in the experiments reported in the original manuscript. We will open-source all the code and ensure support for both FP16 and BF16. As mentioned above, we have also extended FlashAttention to MTLA, and it supports FP16/BF16.
>
> - Question 3:
>     > While resource constraints prevent testing on large-scale LLMs, what evidence suggests this approach will scale effectively to larger models? Could you provide scaling law experiments, even if limited to smaller model sizes?
>     - In fact, our proposed MTLA, like standard attention, still has quadratic complexity, and is therefore fundamentally similar. Since attention has been widely validated for its scaling capabilities, MTLA can naturally be expected to exhibit the same properties. On the other hand, the set of tasks used in the original manuscript is not suitable for conducting scaling law experiments, as models with a very large number of parameters tend to overfit on them. Large-scale LM pretraining tasks are more appropriate for demonstrating scaling effects, since larger models can better exhibit their advantages in such settings with large amount of data. At present, we are indeed unable to afford such large LLM experiments due to limited computational resources.

---

> > ### Comment · Reviewer_xgRJ · 2025-08-06
> >
> > We deeply thank the author for the thorough response, and we are especially pleased to see positive results on the comparison against orthogonal KV pruning methods, as well as the recent FlashAttention-like implementation and its outstanding performance.
> >
> > We are pleased with the other clarifications, and understand that certain large scale ablations are out of reach for the authors due to hardware limitations.
> >
> > We believe these additions strengthen the contribution of the authors, and will update our scores accordingly.

---

### Official Review · Reviewer_2249 · 2025-07-05

**Clarity:** 2
**Significance:** 2
**Originality:** 3
**Rating:** 3
**Confidence:** 4

**Summary:**

This paper introduces Multi-head Temporal Latent Attention (MTLA), a novel attention mechanism that reduces inference-time memory and improves speed by compressing the Key-Value (KV) cache both in the latent and temporal dimensions. To ensure consistency between training and inference behaviors, the paper proposes a stride-aware causal mask.
The evaluation on speech translation, speech recognition, speech understanding, and text summarization shows that MTLA can achieve substantial improvements in efficiency with competitive accuracy.

**Questions:**

See above

**Ethical Concerns:**

["NO or VERY MINOR ethics concerns only"]

**Final Justification:**

After several rounds of discussion with authors and also checking other reviewers' comments,  I think my major concern could not be resolved:

- This time-wise attention compression has been widely studied on attention and this work is an extension on MLA -- so the novelty authors claim is it has not been widely studied on MLA, which is a relatively new variant attention. However, I don't see where the challenges on extending all previously studied attention compression /kv compression mechanism here. Therefore, I think method wise, it is fairly incremental.
- The evaluation and effectiveness of the methods can't be justified IMO with current evaluation and scale.

**Quality:**

3

**Strengths And Weaknesses:**

Strength:
- The paper is well-written and easy to follow.
- The proposed method shows strong empirical performance on speech translation, speech recognition, speech understanding, and text summarization tasks.
- The stride-aware causal mask enables efficient parallel training while preserving inference-time attention behavior

Weakness and Questions:
- Lack of comparison to other token compression works like [1].
- Although the paper mentions FlashAttention in the limitations section, it remains unclear how MTLA compares in real-world end-to-end latency. Integrating MTLA with FlashAttention may not be trivial. It will be great for the paper to have such a kernel to show that the proposed attention can achieve wall-clock time speedup.
- Although the proposed method has been evaluated in various tasks like speech recognition and summarization. I feel that those tasks are not challenging enough. For example, what is the context length for those tasks? Considering that attention only becomes a major bottleneck only when the context length is long enough. It would be great for the paper to also test the proposed methods on more challenging long-context tasks.
- Another concern about the evaluation is that it remains unclear whether the proposed attention mechanism is effective in long-context LLM scenarios, which represent a major challenge for long-context inference. It would strengthen the paper to justify or demonstrate its effectiveness in such settings.

[1] Zhang, Peitian, et al. "Long Context Compression with Activation Beacon." The Thirteenth International Conference on Learning Representations.

---

> ### Author Rebuttal · Authors · 2025-07-30
>
> Thanks for the insightful review: your suggestions will be very helpful for improving the manuscript.
>
> The concerns raised by the reviewer were mentioned in the Limitations section of the original manuscript, such as the engineering effort required to extend FlashAttention to MTLA and validation on more tasks. We initially considered these as future work because we wanted to keep a clear focus of this paper, specifically on presenting our novel method for temporal KV cache compression and validating it on a set of typical tasks. In fact, we have indeed been actively working on these planned future extensions, such as integrating MTLA with FlashAttention. In light of the reviewer's comments, we have decided to include the results of this new extra work here and will also include it in an appendix of the final version of the paper. We hope that this will address the reviewer's concerns and enable the reviewer to revise the overall score.
>
> Below, we provide a point-by-point response to the reviewer's comments.
>
> - Weakness 1:
>     > Lack of comparison to other token compression works like [1].
>     [1] Zhang, Peitian, et al. "Long Context Compression with Activation Beacon." The Thirteenth International Conference on Learning Representations.
>     - Thanks for the suggested reference. As mentioned in the Limitations section, Transformer-based models have been extensively developed by the community over many years, and the number of related techniques precludes an exhaustive comparison. However, the token/context compression work mentioned by the reviewer is indeed relevant and othogonal to our KV cache compression research direction. We will therefore cite and discuss reference [1] in the Related Work section of the final version of the paper.
>
> - Weakness 2:
>     > Although the paper mentions FlashAttention in the limitations section, it remains unclear how MTLA compares in real-world end-to-end latency. Integrating MTLA with FlashAttention may not be trivial. It will be great for the paper to have such a kernel to show that the proposed attention can achieve wall-clock time speedup.
>     - As the reviewer noted, in the Limitations section of the original manuscript we mentioned that extending FlashAttention to MTLA was planned as future work, as it requires substantial engineering effort, while this paper focuses on presenting our novel temporal compression technique. Recently, we have indeed been actively working on extending MTLA, including the integration of FlashAttention. A CUDA kernel based on MTLA has now been implemented. Below, we include some results for MTLA using our extended version of FlashAttention on the same speech translation task presented in Table 1 of the original manuscript.
>         | ST Model  | Quality (BLEU) |Inference Time (s)| Inference GPU Memory (MiB) |
>         | :----------- | :-------: |:-------: | :-------:|
>         |MHA|23.18|281.3|18646|
>         |MHA with FlashAttention|23.16|145.7|9244|
>         |MTLA|23.28|65.6|2835|
>         |MTLA with Extended FlashAttention|23.29|36.5|1259|
>
>         As the results show, using FlashAttention speeds up computation, but it does not change the experimental conclusions. We will include these results and further detail in the appendix of the final version. We will also open-source our extension to FlashAttention for MTLA.
>
> - Weakness 3:
>     > Although the proposed method has been evaluated in various tasks like speech recognition and summarization. I feel that those tasks are not challenging enough. For example, what is the context length for those tasks? Considering that attention only becomes a major bottleneck only when the context length is long enough. It would be great for the paper to also test the proposed methods on more challenging long-context tasks.
>     - This paper has already been evaluated on various long-sequence tasks, since the sequences used for speech tasks and document texts are typically long sequences. For example, in the MuST-C speech translation dataset we used, the maximum sequence length exceeds 5K, with an average length of more than 0.6K; in the XSum dataset, the maximum can be beyond 13K, with an average of more than 0.5K. Hence these tasks would already normally be viewed as long-sequence tasks with suitable training data. In these scenarios, effective handling of the context length becomes the primary bottleneck. This is also why using our proposed MTLA leads to large improvements in inference speed and reductions in GPU memory usage compared to other attention mechanisms. On the other hand, extremely long-sequence tasks often rely on LLMs to exhibit emergent capabilities, since the corresponding training data with extremely long sequences is scarce. As we mentioned in the Limitations section, due to limited resources, we are unable to afford to pretrain LLMs with MTLA, which is why we chose these well-established long-sequence tasks in the original mauscript.
>     - However, in order to further address the reviewer’s concerns, we have additionally conducted evaluations on the Long-Range Arena (LRA) benchmark (Tay et al., 2021). LRA is a widely used benchmark for testing efficient Transformer models in long-context scenarios. We did not include this benchmark in the original version of the paper because LRA is not primarily designed for decoder-only architectures and may be more suitable for encoder self-attention.
>     - Below, we present the results of MTLA on the LRA benchmark (with other results taken from the Tay et al., 2021 paper):
>         | Model| Listops（`↑`）| Text（`↑`） |Retrieval（`↑`）| Image（`↑`） |Pathfinder（`↑`） | Avg（`↑`）|
>         | :----------- | :-------: | :-------: |:-------: |:-------: |:-------: | :-------:|
>         |Transformer| 36.37| 64.27| 57.46| 42.44 |71.40|54.39|
>         |Local Attention| 15.82| 52.98| 53.39| 41.46| 66.63|  46.06|
>         |Sparse Transformers| 17.07 |63.58| 59.59| 44.24| 71.71 | 51.24|
>        | Longformer |35.63| 62.85 |56.89 |42.22| 69.71| 53.46|
>       | Linformer| 35.70 |53.94| 52.27| 38.56| 76.34| 51.36|
>        |Reformer| 37.27 |56.10| 53.40| 38.07| 68.50 |50.67|
>        |Sinkhorn Transformers| 33.67 |61.20 |53.83 |41.23 |67.45 | 51.39|
>        |Synthesizer |36.99| 61.68 |54.67| 41.61 |69.45| 52.88|
>        |BigBird| 36.05 |64.02 |59.29| 40.83| 74.87| 55.01|
>        |Linear Transformer| 16.13| 65.90| 53.09| 42.34| 75.30| 50.55|
>        |Performer |18.01| 65.40| 53.82| 42.77| 77.05|51.41|
>        |**MTLA**|40.47|66.99|59.88|48.10|67.55|**56.60**|
>
>       "ListOps" is a 10-way classification task with a sequence length of 2K. "Text" is a binary classification task with sequences up to 2,048 tokens. "Retrieval" is also a binary classification task, requiring processing of 8K tokens during evaluation. "Image" is a 10-way classification task with a sequence length of 1,024. "Pathfinder" is a binary image classification task with sequences of length 1,024. For more detail about LRA refer to Tay at al. (2021).
>
>     - The results above show that MTLA achieves strong performance on the LRA benchmark compared to other attention mechanisms. Combined with the experiments in the original paper, MTLA consistently demonstrates strong performance across a range of tasks involving text, speech, and vision.
>
>     Tay et al. 2021. "Long Range Arena: A benchmark for efficient transformers." In Proc. ICLR.
>
> - Weakness 4:
>     > Another concern about the evaluation is that it remains unclear whether the proposed attention mechanism is effective in long-context LLM scenarios, which represent a major challenge for long-context inference. It would strengthen the paper to justify or demonstrate its effectiveness in such settings.
>     - We agree that it would further strengthen the paper if we could afford to pretrain an LLM based on MTLA. However, as mentioned in the Limitations section, due to limited computational resources, we are unable to afford to do LLM pretraining, which is why we did not explicitly link MTLA to LLMs in the main text of the original manuscript. This is a challenge we will need to address in future, but we believe the experimental validation we currently provide should be acceptable within the scope of an academic paper. We do plan to actively promote the widespread adoption of MTLA and KV cache temporal compression, so scaling to LLMs is definitely a direction we will strive to investigate in future.
>
>     We hope that our clarifications have effectively addressed your concerns. We noticed that your comments did not raise technical concerns about our proposed method, but rather provided suggestions regarding experimental validation. In the responses above, we have added a substantial number of additional experimental results, which follow your suggestions and provide a broader evaluation of MTLA, demonstrating its effectiveness. We hope you will kindly take into account the new results presented here in your final review scores.

---

> > ### Comment · Reviewer_2249 · 2025-08-04
> > **Additional Questions**
> >
> > I appreciate the authors' efforts with the additional experiments. However, after further examination, I would like to raise several further concerns that I believe are critical to the paper’s technical positioning and evaluation rigor.
> >
> > - Lack of Clear Differentiation from Prior Attention Compression Works: Besides MLTA is built on the top of recent MLA, it remains unclear what core novelty distinguishes MTLA from the extensive literature on token or KV compression on the top of attention. Hundreds of works have explored merging tokens or compressing attention representations, and without a more thorough comparison or ablation against these, it is difficult to assess MTLA’s unique contribution. I'm not very convinced by author's "orthogonal" claim.
> > - Evaluation on Limited Benchmarks: In addition, I took a closer look at the paper's related work session and it claims several weakness of linear atttention methods. However, since MTLA compresses along the time dimension, it is not obvious that this approach provides a principled advantage over well-studied linear attention methods such as Mamba or its predecessors like S4. In fact, MTLA underperforms even S4[1], which is a work around a few year's ago on the LRA benchmark, which questions its competitiveness in long-context modeling. In addition, LRA is also no longer a standard benchmark in the long-context modeling community.
> > - Further question on speed improvements:  What is the performance of MLTA compared to MLA alone (without temporal compression). I'm trying to understand how much speed up was already brought in by MLA which MLTA based on compared to MHA.
> >
> > In conclusion, my main concern with the current version of the paper is that it does not fully reflect the progress in architectural research over the past few years. The choice of benchmarks and baselines leans heavily on very out-dated settings, which limits the ability to fairly assess its contributions in the context of more recent developments. A more comprehensive comparison with up-to-date methods and evaluation on widely adopted modern benchmarks would significantly strengthen the work.

---

> > > ### Author Response · Authors · 2025-08-04
> > > **Following Response to Reviewer 2249 (Part 1 of 3)**
> > >
> > > We’re happy to provide further clarifications. Below, we continue with a point-by-point response to your most recent comments.
> > > - Point 1:
> > >     > Lack of Clear Differentiation from Prior Attention Compression Works: Besides MLTA is built on the top of recent MLA, it remains unclear what core novelty distinguishes MTLA from the extensive literature on token or KV compression on the top of attention. Hundreds of works have explored merging tokens or compressing attention representations, and without a more thorough comparison or ablation against these, it is difficult to assess MTLA’s unique contribution. I'm not very convinced by author's "orthogonal" claim.
> > >     - **Core Novelty:** MTLA is a fundamentally new attention mechanism that leads to temporal compression of the Key-Value (KV) cache. MTLA fully supports parallel training like standard MHA and does not require any further changes beyond the attention module itself. Hence, MTLA can be easily substituted for the self-attention module in any model architecture. To the best of our knowledge, this is the first work of this kind, making it, we believe, an exciting contribution to fundamental research. As Reviewer xgRJ pointed out, this is a promising research direction because there is substantial room for KV cache compression along the temporal dimension.
> > >
> > >     -  **Differences to token compression:**
> > >         - Research related to MTLA includes token pruning/compression methods like SnapKV from our original manuscript and reference [1] suggested by the reviewer. These methods reduce the token sequence length to lower inference cost but usually require changes to inference or training. In contrast, MTLA does not reduce the number of generated tokens but produces shorter KV caches. As a novel type of attention mechanism, MTLA has broad applicability and intuitively can be combined with token pruning/compression methods with minimal changes that would be expected to further reduce the  KV cache size.
> > >         - Additionally, MTLA can be applied to various tasks and modalities, as shown in our manuscript and responses, including speech, text, and vision. Some token compression methods may be less suitable for modalities like speech, which uses continuous representations.
> > >
> > >     - **Further Comparisons:** The Related Work section (lines 67–89) covers some key papers in the areas of KV cache compression and token compression.  We have also compared MTLA to MLA with SnapKV as a representative token compression/pruning method. The results on the same speech translation task presented in Table 1 of the original manuscript are:
> > >       |ST Model|Quality (BLEU)|Inference Time (s)|Inference GPU Memory (MiB)|
> > >       |:-|:-:|:-:|:-:|
> > >       |MLA|22.97|97.0|5065|
> > >       |MLA with SnapKV|21.76|80.8|4222|
> > >       |MTLA|23.28|65.6|2835|
> > >
> > >       The results show that while MLA with SnapKV improves inference efficiency over MLA, it also has some reduction in translation quality.  MTLA outperforms MLA with SnapKV in terms of quality, inference time and GPU memory.
> > >     - **Planned Revisions for Final Version of the Paper:** The final version of the paper will include the comparison of MLA with SnapKV. It will also expand the Related Work section, citing and discussing reference [1] suggested by the reviewer.

---

> > > > ### Author Response · Authors · 2025-08-04
> > > > **Following Response to Reviewer 2249 (Part 2 of 3)**
> > > >
> > > > - Point 2:
> > > >     > Evaluation on Limited Benchmarks: In addition, I took a closer look at the paper's related work session and it claims several weakness of linear atttention methods. However, since MTLA compresses along the time dimension, it is not obvious that this approach provides a principled advantage over well-studied linear attention methods such as Mamba or its predecessors like S4. In fact, MTLA underperforms even S4[1], which is a work around a few year's ago on the LRA benchmark, which questions its competitiveness in long-context modeling. In addition, LRA is also no longer a standard benchmark in the long-context modeling community.
> > > >     - The most important difference between MTLA and linear attention is that MTLA retains quadratic attention, which remains the most widely used approach. While linear attention can indeed offer lower inference cost when dealing with extremely long sequences, model performance often becomes a concern. Some papers, such as reference [2] in our original manuscript, have also pointed out that the quadratic complexity of the attention mechanism is a necessary overhead, although of course alternatives are an interesting area of current research. MTLA reduces the constant in quadratic attention, since it compresses the temporal dimension of the KV cache. When combined with other engineering optimisations, quadratic attention mechanisms can already achieve good inference speed without compromising performance.
> > > >
> > > >     - As a further point of comparison, we additionally implemented Mamba2 and compared it to  MTLA on the same speech translation task presented in Table 1 of the original manuscript. The results are:
> > > >         |ST Model|Quality (BLEU)|Inference Time (s)|Inference GPU Memory (MiB)|
> > > >         |:-|:-:|:-:|:-:|
> > > >         |MHA|23.18|281.3|18646|
> > > >         |Mamba2|18.62|157.5|5676|
> > > >         |MTLA|23.28|65.6|2835|
> > > >
> > > >       The results show that MTLA outperforms Mamba2 in inference efficiency on this task and also on translation quality. While linear-complexity models like Mamba2 will of course yield more efficient inference than quadratic attention mechanisms when dealing with extremely long sequences, the model performance can also suffer. In summary, MTLA follows the mainstream approach of using quadratic attention mechanisms and therefore benefits from stronger model performance while greatly improving inference time and GPU memory usage.
> > > >   - Regarding the performance of S4 on LRA, it is known that State Space Models (SSMs), including S4, perform better than Transformer attention on the LRA benchmark, which is particularly challenging for attention mechanisms. SSMs can be reformulated as long convolutions with kernel sizes matching the full sequence length. The tasks in LRA involve long contexts but also possess a strong positional nature, which strongly favors long-convolution-based architectures with a time-decay mechanisms. We chose the LRA benchmark because it is very challenging and was originally designed to evaluate attention mechanisms. Hence we aimed to validate our approach on more difficult tasks to address the reviewer’s concerns. Therefore, we focused on a comparison with various attention variants on the LRA benchmark. We believe that the results on the LRA benchmark can still provide useful complementary information.

---

> > > > > ### Author Response · Authors · 2025-08-04
> > > > > **Following Response to Reviewer 2249 (Part 3 of 3)**
> > > > >
> > > > > - Point 3:
> > > > >     > Further question on speed improvements: What is the performance of MLTA compared to MLA alone (without temporal compression). I'm trying to understand how much speed up was already brought in by MLA which MLTA based on compared to MHA.
> > > > >     - In addition to the comparisons in the original paper, we present below a comparison between MTLA, MLA and MHA, both with and without FlashAttention.  The results on the same speech translation task presented in Table 1 of the original manuscript are as follows:
> > > > >         |ST Model|Quality (BLEU)|Inference Time (s)|Inference GPU Memory (MiB)|
> > > > >         |:-|:-:|:-:|:-:|
> > > > >         |MHA|23.18|281.3|18646|
> > > > >         |MHA with FlashAttention|23.16|145.7|9244|
> > > > >         |MLA|22.97|97.0|5065|
> > > > >         |MLA with Extended FlashAttention|22.98|50.0|2434|
> > > > >         |MTLA|23.28|65.6|2835|
> > > > >         |MTLA with Extended FlashAttention|23.29|36.5|1259|
> > > > >
> > > > >         The results show that MLA brings large speedups over MHA, but the speedup of MTLA over MLA is also substantial.
> > > > >
> > > > > - Point 4:
> > > > >     > In conclusion, my main concern with the current version of the paper is that it does not fully reflect the progress in architectural research over the past few years. The choice of benchmarks and baselines leans heavily on very out-dated settings, which limits the ability to fairly assess its contributions in the context of more recent developments. A more comprehensive comparison with up-to-date methods and evaluation on widely adopted modern benchmarks would significantly strengthen the work.
> > > > >     - We compared MTLA to MLA throughout the paper since MLA is one of the most advanced KV cache architectures and first appeared  in the DeepSeek-V2 paper on arXiv only a year before our NeurIPS submission. We believe that our experimental setup is also up to date. For example, in the speech recognition task, we use self-supervised speech representations as input features to build a baseline with very good performsnce given the amount of labelled speech training data. In addition, during the response stage, we included comparisons with advanced methods such SnapKV, Mamba2 as well as extending FlashAttention to be applicable to MTLA. We would welcome any further detailed suggestions from the reviewer which would enable us to further improve the final version of our paper.
> > > > >     - Although we are not entirely sure what the reviewer refers to by "modern benchmarks", we would like to emphasise that this paper is not an LLM-focused study, and we do not have the resources to support LLM pretraining. Even at the scale of models used in our experiments, our method already achieves large reductions in both inference time and GPU memory, which the reviewer has acknowledged. While our method has clear potential to be applied to LLMs, that is not the focus of this paper and is left to further work.
> > > > >
> > > > > We hope that our clarifications have effectively answered your questions. We hope you will kindly take into account our contributions in your final review scores. Our proposed MTLA technique is a novel attention mechanism, which is an exciting research contribution that has been acknowledged by other reviewers. As shown in our response, extensions such as the FlashAttention variant, although complex as noted by the reviewer, will provide meaningful contributions to the community. To support this, we will fully open-source our code.

---

> > > > ### Comment · Reviewer_2249 · 2025-08-04
> > > > **What's the difference between temporal and token dimension**
> > > >
> > > > I'm still a bit unclear about the authors' novelty claims regarding:
> > > >
> > > > (1) Being the first (or among the first) to compress the temporal dimension for long sequences. If I understand correctly, most of the earlier works—such as those using RNNs, SSMs, convolutions, token merging approaches in newer architectures all perform some form of temporal compression. So basically almost any paper that is not compressing the dimension d is compressing temporal dimension. It would be helpful if the authors could help me understand **what's new in temporal compression** and what distinguishes their method from these prior approaches. If not, I maintain my original concern that the paper includes too few baselines and makes claims that may be somewhat overstated.
> > > >
> > > > (2) The evaluation setup, particularly using benchmarks that are now widely acknowledged to be insufficient for testing long-sequence capabilities. As is commonly recognized, SSMs tend to perform well on these benchmarks, but the datasets in question are not generally considered strong indicators of true long-range dependency. Therefore I'm not very convinced of the additional results.
> > > >
> > > > (3) I believe summarization and speed are both strong application areas for SSMs. If the evaluation focuses primarily on these tasks, it becomes difficult to isolate the benefits of constant compression in attention mechanisms. Given that Mamba-based models are known to be sensitive to initialization, could the authors provide more details on the experimental setup and any measures taken to ensure fair comparisons? While SSMs may underperform in tasks requiring true long-range dependency, they are generally competitive—and often superior—in summarization tasks. Some clarification on this point would be appreciated.

---

> > > > > ### Author Response · Authors · 2025-08-05
> > > > > **Further Response to Reviewer 2249 (Part 1 of 2)**
> > > > >
> > > > > We thank the reviewer for actively engaging in this stage of the discussion. We do note that we have already responded to a number of concerns raised by the reviewer which include the performance of MTLA when using FlashAttention and the differences between KV cache temporal compression and token merging.
> > > > >
> > > > > We'd like to continue to provide clarifications in response to the reviewer’s most recent questions. Below, we proceed with point-by-point responses.
> > > > >
> > > > > - Point 1:
> > > > >     - First, regarding the reviewer's concern about novelty of temporal compression, we would like to emphasise that **our contribution lies in compressing the temporal dimension of the self-attention KV cache**. This is explicitly stated in the abstract and the contributions listed in the Introduction of the submitted manuscript (lines 57-65). We are not making any broader claims regarding temporal compression.
> > > > >   - MTLA is a new attention mechanism that reduces the amount of  computation and memory use for the KV cache as used in transformer decoder architectures. MTLA preserves all of the core properties of attention, including the quadratic computational complexity, which, as mentioned in our previous response, is considered a necessary overhead. Furthermore, MTLA does not require any changes beyond the attention module itself, making it lightweight and easy to use in place of standard multi-head attention.
> > > > >   - Other paradigms for sequence modelling, such as RNNs, clearly perform temporal modelling and in RNNs there is explicit temporal compression, but the novelty of MTLA is that we have applied temporal compression  to the KV cache. It is possible that the reviewer is thinking that other  sequence modelling paradigms could be applied to the KV cache temporal compression problem and while this may be case, there are clear constraints on temporal modelling in this context in order to retain the parallel training property of transformers, the plug-in advantages of the MTLA scheme, and the overall task performance of the model.
> > > > >   -  Compared to compressing the token dimension for the KV cache as in MLA, temporal compression for the KV cache is more challenging, since the temporal length is different for each sample. Therefore a temporal compression scheme needs to be designed to allow parallel training that also matches the inference behaviour with a temporally-compressed KV cache.
> > > > >    - We respect the research efforts exploring alternatives to attention, particularly SSMs. However, it is clear that we are not in a position to determine whether SSMs or attention mechanisms are ultimately preferable for particular tasks. Currently, attention with quadratic complexity remains the mainstream choice, and our work aims to contribute to this widely adopted module class.
> > > > >     - Regarding the comparison with token merging, we kindly invite the reviewer to refer again to our previous response titled "Following Response to Reviewer 2249 (Part 1 of 3)." We briefly summarise the key points again below.
> > > > >         - Token merging usually requires changes to inference or training, while MTLA is a new attention mechanism that does not require any changes beyond the attention module itself. This is a critical advantage for real-world deployment.
> > > > >         - As an attention mechanism, MTLA can be easily applied to various tasks and modalities, as shown in our manuscript and responses, including speech, text, and vision. In contrast, some token compression methods may be less suitable for modalities such as speech, which involve continuous representations (i.e., without explicit tokens).
> > > > >         - MTLA does not reduce the number of generated tokens but instead produces shorter KV caches. This is why we previously mentioned that MTLA could potentially be combined with token merging.
> > > > >
> > > > >   Finally we note that temporal compression of the KV cache for a transformer decoder is indeed a challenge, as the model generates tokens one by one during inference. After temporal compression of the KV cache, adjacent generation steps have caches of the same length but containing different information, which makes parallel training difficult. Our proposed MTLA approach elegantly addresses this issue through the use of a stride-aware causal mask, while preserving the simplicity of the attention mechanism. We believe that the large reductions in computation and GPU memory use compared to both MHA and MLA means that this is an important contribution.

---

> > > > > > ### Author Response · Authors · 2025-08-05
> > > > > > **Further Response to Reviewer 2249 (Part 2 of 2)**
> > > > > >
> > > > > > - Point 2:
> > > > > >     > The evaluation setup, particularly using benchmarks that are now widely acknowledged to be insufficient for testing long-sequence capabilities. As is commonly recognized, SSMs tend to perform well on these benchmarks, but the datasets in question are not generally considered strong indicators of true long-range dependency. Therefore I'm not very convinced of the additional results.
> > > > > >     - As we mentioned in our previous response, the results on the LRA benchmark provided during the response stage were intended to offer supplemental information and to cover a broader range of tasks. As noted in Reviewer yMqX's recent comment, the LRA results are indeed useful. In fact, the speech tasks evaluated in our original manuscript are also challenging. For example, speech translation involves long speech sequences and the translation task additionally requires output sequence reordering. This poses a clear test of whether a model can retain enough useful information while taking account of a significant amount of context. As we previously demonstrated, the use of SnapKV incurred some performance degradation on this task.
> > > > > >     - We note the reviewer's views on the benchmarks we are using. We therefore wonder if the reviewer has any specific suggestions for evaluation tasks/data sets that we should use given the type of models that we are using and our computational constraints, which mean that we have been unable to apply MTLA to LLMs.
> > > > > >
> > > > > > - Point 3:
> > > > > >     > I believe summarization and speed are both strong application areas for SSMs. If the evaluation focuses primarily on these tasks, it becomes difficult to isolate the benefits of constant compression in attention mechanisms. Given that Mamba-based models are known to be sensitive to initialization, could the authors provide more details on the experimental setup and any measures taken to ensure fair comparisons? While SSMs may underperform in tasks requiring true long-range dependency, they are generally competitive—and often superior—in summarization tasks. Some clarification on this point would be appreciated.
> > > > > >     - For constructing the Mamba2 model, we used the official Mamba codebase and followed its implementation. We built the Mamba2 model using the parameters recommended in the README of the official Mamba repository and, as suggested, set the number of Mamba2 layers to be twice that of the corresponding Transformer layers. The final number of parameters is comparable to that of our Transformer models, with the Mamba2 model having approximately 1 million more parameters. We would like to emphasise that comparing attention mechanisms with SSMs is definitely not the focus of this paper. This is a very broad topic that requires collective efforts from the community across different tasks and applications. This paper simply follows the mainstream attention-based models and aims to introduce some improvements within that framework.

---

> > > > > > > ### Comment · Reviewer_2249 · 2025-08-08
> > > > > > > **Summary**
> > > > > > >
> > > > > > > Thanks for all the detailed clarification and response. I think my major concern could not be resolved. In summary, the functionality of this work is an extension of time-wise attention compression study to the MLA variant, so the problem is not very new. The methodological contribution appears incremental too — given that existing compression techniques (e.g., KV or attention compression mechanisms) could plausibly be adapted to MLA without substantial modification and there're lack of strong baseline comparisons or strong justification on what are special challenges in MLA. Additionally, the current evaluation, in terms of scale and experimental scope, does not convincingly demonstrate the method's efficacy or broader applicability. Therefore, I decide to keep my original rating.

---

> > > > > > > > ### Author Response · Authors · 2025-08-09
> > > > > > > >
> > > > > > > > We thank the reviewer for their continued engagement in the discussion. We note the major issue that the reviewer has now raised is that "this work is an extension of time-wise attention compression study to the MLA variant, so the problem is not very new". We have previously discussed the differences between our proposed MTLA method and previous work on token/context compression (see Further Response to Reviewer 2249, Part 1 of 2).
> > > > > > > >
> > > > > > > > We would like to make the following comments.
> > > > > > > >
> > > > > > > > - The main contribution of this paper lies in the temporal compression of the attention KV cache, with MLA serving as a strong baseline. In our earlier response (i.e. Following Response to Reviewer 2249, Part 1 of 3), we also provided the results of applying SnapKV to MLA and compared it with our proposed MTLA technique. In both the original manuscript and our earlier responses, we have emphasised that temporal compression is challenging because it requires achieving parallel training while matching the behavior during incremental inference. This was specifically illustrated in Figure 2 of the original manuscript.
> > > > > > > >
> > > > > > > > - Regarding the time-wise attention compression study, could the reviewer kindly provide some references? Do these methods, like our MTLA mechanism, support KV cache temporal compression without requiring any changes beyond the attention module itself? We are not aware of any such previous work as we previously mentioned, KV cache temporal compression is performed during incremental inference while also preserving the parallel training properties of attention, which is challenging. This was explicitly illustrated in Figure 2(b) of our original paper.
> > > > > > > >
> > > > > > > >
> > > > > > > > - Regarding the scale and scope of the evaluation, we have already mentioned that we are indeed limited by computational resources and cannot afford LLM pretraining, which other reviewers have all understood. Therefore, in our earlier response (i.e., Further Response to Reviewer 2249, Part 2 of 2), we asked that the reviewer to kindly provide specific suggestions on which benchmarks would be most appropriate for our situation, so that we could conduct experiments to address their concerns. However, the reviewer did not respond with any specific suggestions. We would greatly appreciate specific comments that can help us further improve the manuscript.
> > > > > > > >
> > > > > > > > - During the rebuttal process, we have tried to clarify the novelty of our approach and provided additional experimental results. We believe we have addressed several of the reviewer’s concerns, including the extension to FlashAttention, which, as reviewer xgRJ noted, represents a substantial effort. The reviewer has raised several new points during the discussion and we have still tried to address the new points raised.
> > > > > > > >
> > > > > > > > We have tried to clarify all of the issues raised by the reviewer and have provided significant number of new results. We remain encouraged by the positive feedback from all the other reviewers and believe that our work is an important contribution for the improved inference efficiency of decoder-only transformer models.

---

### Decision · Program_Chairs · 2025-09-17

**Decision:**

Accept (poster)

**Comment:**

This paper introduces Multi-head Temporal Latent Attention (MTLA), a technique for improving the inference efficiency of Transformer models. MTLA compresses the Key-Value (KV) cache along the temporal dimension by dynamically merging adjacent cache vectors using a hyper-network. This method extends the recently proposed Multi-head Latent Attention (MLA), which compresses the cache along the feature dimension, thus achieving compression in both dimensions. The authors also propose a stride-aware causal mask to ensure consistency between parallel training and incremental inference.

*   Strengths: The paper focuses on the critical and well-motivated problem of KV cache memory growth in autoregressive models. The core idea of learnable temporal compression of the KV cache is an important and timely research direction (xgRJ, hHHE), and the setup building on MLA work is interesting and well-motivated. The paper is well written and easy to follow (hHHE) and the proposed stride-aware casual mask is a simple and effective solution. The method demonstrates good empirical improvements in inference speed and memory reduction across a variety of tasks without sacrificing performance (2249, yMqX).

*   Weaknesses: The two main weaknesses of the paper are (1) having experiments only on very small scale models (only 100M params). Even with limited resources these days this is considered very small scale, and multiple reviewers found it unconvincing since it is unclear how MTLA will perform for larger more practical models (hHHE, xgRJ, yMqX). Also (2) reviewers brought up some concerns about the positioning of the paper and the baselines it is comparing against (​​2249, xgRJ). The paper is motivated and positioned as an extension of MLA, which is logical from the method point of view, but due to the goal of the method which unlike MLA focuses on reducing the length of KV, there are multiple techniques for KV pruning or efficient attention that should be compared against as baselines. During the rebuttal the authors added some comparison to SnapKV and Mamba2. Further comparisons will strengthen the paper.